# Caspase-11 counteracts mitochondrial ROS-mediated clearance of *Staphylococcus aureus* in macrophages

Kathrin Krause[1], Kylene Daily[1], Shady Estfanous[1], Kaitlin Hamilton[1], Asmaa Badr[1], Arwa Abu Khweek[1,2], Rana Hegazi[1], Midhun NK Anne[1], Brett Klamer[3], Xiaoli Zhang[3], Mikhail A Gavrilin[4], Vijay Pancholi[5] & Amal O Amer[1,*] (iD)

## Abstract

Methicillin-resistant *Staphylococcus aureus* (MRSA) is a growing health concern due to increasing resistance to antibiotics. As a facultative intracellular pathogen, MRSA is capable of persisting within professional phagocytes including macrophages. Here, we identify a role for CASP11 in facilitating MRSA survival within murine macrophages. We show that MRSA actively prevents the recruitment of mitochondria to the vicinity of the vacuoles they reside in to avoid intracellular demise. This process requires CASP11 since its deficiency allows increased association of MRSA-containing vacuoles with mitochondria. The induction of mitochondrial superoxide by antimycin A (Ant A) improves MRSA eradication in $casp11^{-/-}$ cells, where mitochondria remain in the vicinity of the bacterium. In WT macrophages, Ant A does not affect MRSA persistence. When mitochondrial dissociation is prevented by the actin depolymerizing agent cytochalasin D, Ant A effectively reduces MRSA numbers. Moreover, the absence of CASP11 leads to reduced cleavage of CASP1, IL-1β, and CASP7, as well as to reduced production of CXCL1/KC. Our study provides a new role for CASP11 in promoting the persistence of Gram-positive bacteria.

Keywords caspase-11; macrophages; mitochondria; ROS; *Staphylococcus aureus*
Subject Categories Autophagy & Cell Death; Immunology; Microbiology, Virology & Host Pathogen Interaction

## Introduction

Methicillin-resistant *Staphylococcus aureus* (MRSA) refers to a group of Gram-positive cocci that have developed a resistance to most β-lactam antibiotics due to the expression of a penicillin-binding protein (PBP2a) [1]. As an opportunistic pathogen, *S. aureus* exhibits a broad repertoire of virulence factors and can cause a variety of clinical manifestations, ranging from localized mild skin and soft tissue infections to severe invasive diseases with potentially fatal outcomes such as pneumonia, endocarditis, and sepsis [2,3]. Genetically diverse MRSA isolates can be found in healthcare facilities as well as communities all over the world, and resistances against antibiotics of last resort, such as vancomycin, have emerged [4]. Alternative treatment strategies are therefore necessary to overcome multidrug-resistant MRSA infections.

Inflammatory caspase-11/caspase-4 (CASP11) contributes to non-canonical NLRP3 inflammasome activation and subsequent inflammation [5]. CASP11 is not expressed in healthy tissue unless induced by infection or other pathologic stress [6–9]. Until recently, appreciated functions of CASP11 were the recognition of cytosolic LPS followed by the activation of CASP1, cleavage of gasdermin D (GSDMD), pro-inflammatory cytokine secretion, and cell death [5,9–11]. Additionally, the role of CASP11 is dependent on the infectious agent. While CASP11 deficiency has been shown to protect mice from LPS-induced endotoxemia due to reduced release of the inflammatory mediators IL-1α, IL-1β, and CXCL1/KC [5,9,12], the absence of CASP11 in the context of Gram-negative bacterial infections promotes bacterial replication and dissemination in mice [8,9,13,14]. Furthermore, CASP11 was shown to modulate the intracellular trafficking of pathogens, such as *L. pneumophila* and *B. cenocepacia,* leading to their degradation within lysosomes [8,9,13]. In contrast, little is known about the role of CASP11 in the immune defense against Gram-positive bacteria. Recently, purified lipoteichoic acid (LTA), a cell wall component from Gram-positive bacteria, was reported to induce CASP11 activity via NLRP6 [15]. However, unlike mice infected with Gram-negative bacteria, mice deficient of CASP11 exhibit improved survival and efficient bacterial clearance in response to

1   Department of Microbial Infection and Immunity, Infectious Diseases Institute, Ohio State University, Columbus, OH, USA
2   Department of Biology and Biochemistry, Birzeit University, Birzeit, West Bank, Palestine
3   Center for Biostatistics, Ohio State University, Columbus, OH, USA
4   Department of Internal Medicine, Ohio State University, Columbus, OH, USA
5   Department of Pathology, Ohio State University, Columbus, OH, USA
    *Corresponding author. Tel: +1 614 247 1566; Fax: +1 614 292 9616; E-mail: amal.amer@osumc.edu

Gram-positive pathogens such as *Listeria monocytogenes* and *S. aureus* [15]. The study by Hara *et al* demonstrated that increased production of IL-18 in WT mice impairs clearance of *L. monocytogenes*. However, during *S. aureus* infection, others have shown that the neutralization of IL-1β or IL-18 does not influence survival or pulmonary burdens of mice [16]. Therefore, the mechanism behind reduced susceptibility of *casp11*$^{-/-}$ mice to Gram-positive bacteria is not completely understood and may vary according to the pathogen.

Mitochondria are the main source for cellular ATP production. While mitochondrial reactive oxygen species (mtROS), such as superoxide and hydrogen peroxide, are generally considered byproducts of oxidative phosphorylation at the inner mitochondrial membrane, there is increasing evidence that mtROS can also augment the bactericidal capacity of macrophages [17,18]. The recruitment of mitochondria to phagosomes containing intracellular bacteria and subsequent elevated mtROS production have been shown to mediate the antibacterial response against *Salmonella enterica* serovar Typhimurium [17]. Likewise, TNF-induced mtROS facilitate clearance of *Mycobacterium tuberculosis* from macrophages [19]. Co-localization of internalized *S. aureus* with mitochondria was documented for both α-hemolysin (Hla)-deficient strains and in response to chemical inhibition of NLRP3, resulting in bacterial clearance by mtROS [16]. Here, we propose a role for CASP11 in facilitating MRSA evasion from mtROS-mediated killing. We report that CASP11 deficiency leads to an increased association of MRSA with mitochondria, which is accompanied by elevated mtROS production and decreased inflammasome activation, thereby promoting more efficient clearance from murine macrophages. Antimycin A (Ant A) treatment, which inhibits complex III of the electron transport chain (ETC) thus raising mitochondrial superoxide production, further improves the bactericidal capacity of *casp11*$^{-/-}$ macrophages against MRSA. In WT macrophages, the inhibition of the actin cytoskeleton via cytochalasin D (Cyto D) prevents the dissociation of phagocytosed MRSA from mitochondria and hence restores Ant A-induced bacterial killing. Together, these results provide a novel role for CASP11 in the pathogenesis of Gram-positive bacteria.

## Results

### CASP11 contributes to MRSA-induced inflammasome activation in murine macrophages

*Staphylococcus aureus* activates CASP1 through the NLRP3 inflammasome, leading to the secretion of IL-1β and cell death [20–23]. While CASP11 was long believed to solely recognize cytosolic LPS from Gram-negative bacteria, leading to non-canonical NLRP3

inflammasome activation [6,7], LTA derived from Gram-positive bacteria has been shown to promote CASP11 cleavage and activation [15]. Since resting cells exhibit low levels of CASP11, we infected bone marrow-derived macrophages (BMDMs) from WT, *casp11*$^{-/-}$, and *casp1*$^{-/-}$ mice with the MRSA strain USA300 at MOI 5:1 to determine whether intracellular MRSA stimulates CASP11 protein expression. At 24 h post-infection, we found a significant up-regulation of CASP11 in cell lysates of infected WT as well as *casp1*$^{-/-}$ macrophages (Fig 1A). To further elucidate whether CASP11 plays a role in inflammasome activation in response to MRSA, we evaluated cleavage of CASP1 and IL-1β in cell culture supernatants from WT and *casp11*$^{-/-}$ macrophages via Western blot analysis at 24 h post-infection (MOI 20:1). BMDMs from *casp1*$^{-/-}$ and *gsdmd*$^{-/-}$ mice were included to serve as a control for absent CASP1 or IL-1β secretion, respectively. In addition, we also added BMDMs deficient of the pseudokinase MLKL, which is a known effector protein for the necroptosis pathway. The inhibition of MLKL was demonstrated to reduce MRSA-induced IL-1β secretion from THP-1 macrophages [24], suggesting potential crosstalk between pyroptosis and necroptosis during MRSA infection. Compared to corresponding WT cells, cleavage and secretion of CASP1 and IL-1β were significantly reduced in supernatants from MRSA-infected *casp11*$^{-/-}$, *gsdmd*$^{-/-}$, and *casp1*$^{-/-}$ but not *mlkl*$^{-/-}$ macrophages (Fig 1B and C). Since CASP1 has been shown to activate CASP7 [25,26], we also analyzed the processing of CASP7 in response to MRSA. Similar to CASP1 and IL-1β, we found significantly lower amounts of CASP7 cleavage products in the supernatants from infected *casp11*$^{-/-}$, *gsdmd*$^{-/-}$, and *casp1*$^{-/-}$ but not *mlkl*$^{-/-}$ macrophages (Fig EV1A). In accordance with decreased secretion of CASP1, IL-1β, and CASP7, there were also lower amounts of the enzyme lactate dehydrogenase (LDH) in cell culture supernatants from MRSA-infected *casp11*$^{-/-}$, *gsdmd*$^{-/-}$, and *casp1*$^{-/-}$ macrophages when compared to WT or *mlkl*$^{-/-}$ cells (Fig EV1B). Interestingly, *gsdmd*$^{-/-}$ macrophages demonstrated mildly increased levels of secreted IL-1β and CASP7 compared to their *casp11*$^{-/-}$ and *casp1*$^{-/-}$ counterparts (Fig 1B), suggesting GSDMD-independent mechanisms contributing to the secretion of cleaved IL-1β and CASP7. Reduced secretion of IL-1α and CXCL1/KC was found only in *casp11*$^{-/-}$ macrophages (Fig 1C). We previously reported decreased CXCL1/KC production in *casp11*$^{-/-}$ macrophages and mice in response to *B. cenocepacia* infection [9]. Here, our data with MRSA support the idea of a general defect in CXCL1/KC production in the absence of CASP11. No difference in the secretion of the inflammasome independent cytokine TNF could be observed among all five macrophage genotypes. Together, these results demonstrate that CASP11 but not MLKL contributes to MRSA-induced activation of CASP1 and subsequent processing of IL-1β and CASP7 in murine macrophages.

**Figure 1. CASP11 contributes to MRSA-induced inflammasome activation in murine macrophages.**

A   Immunoblot analysis of CASP11 from WT, *casp11*$^{-/-}$, and *casp1*$^{-/-}$ BMDMs infected with MRSA (MOI 5:1) at 24 h post-infection. Densitometry analysis represents mean ± SEM (*n* = 3 biological replicates). Statistical analysis was performed using two-way ANOVA. **$P \leq 0.01$, NT = no treatment.

B   Immunoblot analysis of cleaved CASP1 and IL-1β in supernatants from WT, *casp11*$^{-/-}$, *gsdmd*$^{-/-}$, *casp1*$^{-/-}$, and *mlkl*$^{-/-}$ BMDMs infected with MRSA (MOI 20:1) at 24 h post-infection. Densitometry analysis represents mean ± SEM (*n* = 5 biological replicates). Statistical analysis was performed using two-way ANOVA. *$P \leq 0.05$, **$P \leq 0.01$, ***$P \leq 0.001$, NT = no treatment.

C   Cytokine release from WT, *casp11*$^{-/-}$, *gsdmd*$^{-/-}$, *casp1*$^{-/-}$, and *mlkl*$^{-/-}$ BMDMs infected with MRSA (MOI 20:1) at 24 h post-infection. Data represent mean ± SEM (*n* = 8 biological replicates). Statistical analysis was performed using one-way ANOVA. **$P \leq 0.01$.

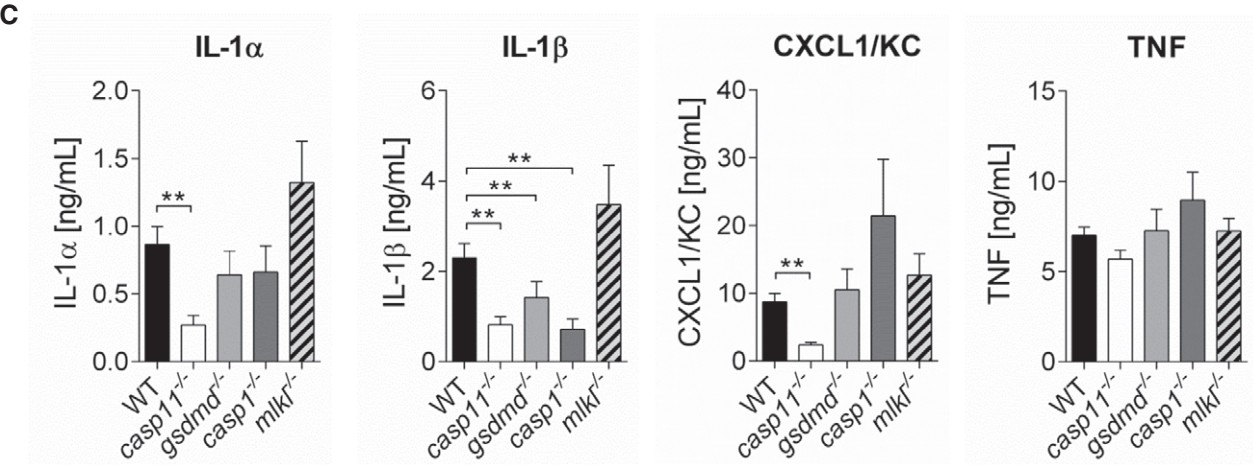

**Figure 1.**

## Human CASP4 and CASP5, homologs of murine CASP11, are involved in IL-1β secretion and cell death induced by MRSA in THP-1 macrophages

To determine whether CASP4 and CASP5, the human homologs of murine CASP11, contribute to MRSA-induced inflammasome activity in human THP-1 macrophages, we used corresponding knockout cells generated using the CRISPR/Cas system (a generous gift of Dr. Seth Masters, Walter and Eliza Hall Institute of Medical Research, Australia) [27]. Single $CASP1^{-/-}$ and Cas9 THP-1 cells (from Dr. Masters) were included as controls. The CRISPR system was induced by incubating the cells with doxycycline. Then, THP-1 cells were differentiated into macrophages using phorbol 12-myristate 13-acetate (PMA) and infected with MRSA at MOI 5:1 for 24 h. As compared to Cas9 control cells, cleavage products of IL-1β and CASP1 were significantly reduced in the supernatants of infected $CASP4/5^{-/-}$ cells (Fig 2A). $CASP4/5^{-/-}$ cells also exhibited reduced processing of pro-CASP1 in cell lysates (Fig 2B) and a decreased release of LDH in response to MRSA (Fig 2C). In addition, the absence of both CASP4 and CASP5 resulted in lower amounts of IL-1α and IL-8 (human homolog of murine CXCL1/KC) in the supernatants of infected cells (Fig 2D). Similar to murine macrophages, we observed no difference in the release of TNF among all 5 genotypes. These results thus indicate that, similar to murine CASP11, both human CASP4 and CASP5 contribute to MRSA-induced cleavage of CASP1 and IL-1β and subsequent cell death.

## CASP11 deficiency improves clearance of intracellular MRSA

Although earlier reports indicated that CASP11 has no role in protection against *L. monocytogenes* infection [28], $casp11^{-/-}$ mice were recently demonstrated to have improved survival rates and lower bacterial loads in response to intravenously delivered *L. monocytogenes* as well as *S. aureus* 8325-4 [15], indicating that the absence of CASP11 helps with the clearance of some Gram-positive bacteria. To investigate the impact of CASP11 on bacterial pulmonary loads, we intratracheally infected WT and $casp11^{-/-}$ mice with $2.5 \times 10^8$ CFU of MRSA and determined the bacterial burden at 96 h post-infection. We found significantly reduced numbers of MRSA in the lungs of $casp11^{-/-}$ mice as compared to corresponding WT mice (Fig 3A). Since MRSA was also shown to invade and persist in murine and human macrophages [29,30], we next elucidated the role of CASP11 for the intracellular survival of MRSA. BMDMs from WT, $casp11^{-/-}$, and $casp1^{-/-}$ mice were infected with MRSA, and intracellular bacterial numbers were determined at 2.5 and 24 h post-infection.

As shown in Fig 3B, the intracellular load of MRSA was significantly lower at 24 h compared to 2.5 h for all three genotypes, indicating bacterial clearance. However, compared to WT or $casp1^{-/-}$ cells, the reduction of intracellular MRSA was more pronounced in macrophages lacking CASP11. LDH release was similar between all three genotypes (Fig 3C). No difference in the intracellular burden of MRSA in WT and $casp11^{-/-}$ macrophages could be observed at earlier time points (Fig EV1C and D). Testing different MOIs of 0.5:1 and 20:1 revealed increased restriction of MRSA at 24 h post-infection in the absence of CASP11 as well (Fig EV1E). In contrast, while the inhibition of NLRP3 was reported to promote clearance of MRSA from human macrophages [16], we did not find improved killing of MRSA in $nlrp3^{-/-}$ BMDMs (Fig 3D). Since GSDMD is directly cleaved by CASP11, we also determined intracellular survival of MRSA in $gsdmd^{-/-}$ macrophages. Yet, no difference in intracellular bacterial numbers could be found between WT and $gsdmd^{-/-}$ macrophages (Fig 3E). Together, these *in vivo* and *in vitro* data suggest that the macrophage response against MRSA is more effective in the absence of CASP11.

## CASP11 deficiency facilitates MRSA clearance through mtROS

It has been shown that mitochondrial ROS production drives the bactericidal activity of macrophages against MRSA [31] and that Hla-induced NLRP3 inflammasome activation negatively affects the association of phagosomes containing MRSA with mitochondria [16]. To test whether mitochondria play a role in the clearance of MRSA in the absence of CASP11, we analyzed co-localization events between MRSA and mitochondria either stained with MitoTracker Deep Red or an anti-Tom20 antibody in WT and $casp11^{-/-}$ macrophages. As indicated in Fig 4A–D, $casp11^{-/-}$ macrophages display a higher percentage of bacteria co-localized with MitoTracker or Tom20 than corresponding WT cells. Electron microscopy revealed bacteria in proximity to mitochondria in $casp11^{-/-}$ but not WT macrophages (Fig 4E). Additionally, to elucidate whether only physical proximity to the mitochondria allows bacterial killing or whether mtROS production itself is increased in $casp11^{-/-}$ macrophages, we measured mitochondrial superoxide production using the MitoSOX Red reagent. Ant A, an inhibitor of complex III of the mitochondrial ETC, served as positive control for superoxide induction, and either the general antioxidant N-acetylcysteine (NAC) or the mtROS-quenching agent MitoQ was used to inhibit mtROS production. The chemicals had no direct bactericidal activity against MRSA (Fig EV2A and B). Compared to non-infected control cells, MRSA suppressed

---

**Figure 2. CASP4/5 mediate MRSA-induced CASP1 activation and IL-1β release in human THP-1 macrophages.**

A  Immunoblot analysis of cleaved IL-1β and CASP1 (p20) in supernatants from Cas9 control, $CASP4^{-/-}$, $CASP5^{-/-}$, $CASP4/5^{-/-}$, and $CASP1^{-/-}$ THP-1 cells infected with MRSA at 24 h post-infection. Densitometry analysis represents mean ± SEM ($n = 4$ biological replicates). Statistical analysis was performed using two-way ANOVA. \*\*$P \leq 0.01$, \*\*\*$P \leq 0.001$, NT = no treatment.

B  Immunoblot analysis of pro-CASP1 (p45) in lysates from Cas9 control, $CASP4^{-/-}$, $CASP5^{-/-}$, $CASP4/5^{-/-}$, and $CASP1^{-/-}$ THP-1 cells infected with MRSA at 24 h post-infection. Densitometry analysis represents mean ± SEM ($n = 5$ biological replicates). Statistical analysis was performed using two-way ANOVA. \*$P \leq 0.05$, \*\*\*$P \leq 0.001$, NT = no treatment.

C  MRSA-induced LDH released in supernatants from Cas9 control, $CASP4^{-/-}$, $CASP5^{-/-}$, $CASP4/5^{-/-}$, and $CASP1^{-/-}$ THP-1 cells at 24 h post-infection. Data represent mean ± SEM ($n = 15$ biological replicates). Statistical analysis was performed using one-way ANOVA. \*$P \leq 0.05$, \*\*\*$P \leq 0.001$.

D  Cytokine release from Cas9 control, $CASP4^{-/-}$, $CASP5^{-/-}$, $CASP4/5^{-/-}$, and $CASP1^{-/-}$ THP-1 cells infected with MRSA at 24 h post-infection. Data represent mean ± SEM ($n = 10$ biological replicates). Statistical analysis was performed using one-way ANOVA. \*$P \leq 0.05$, \*\*$P \leq 0.01$.

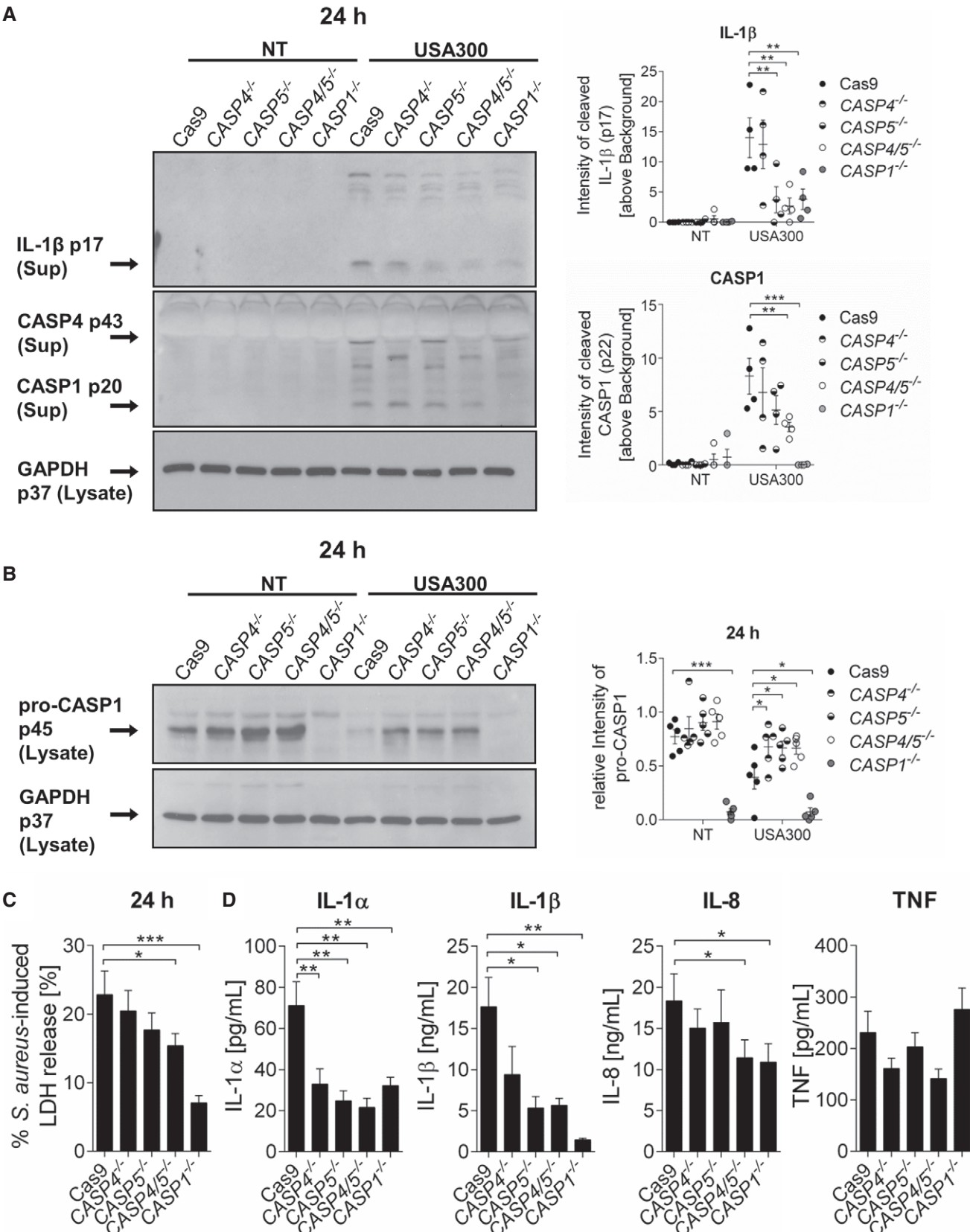

**Figure 2.**

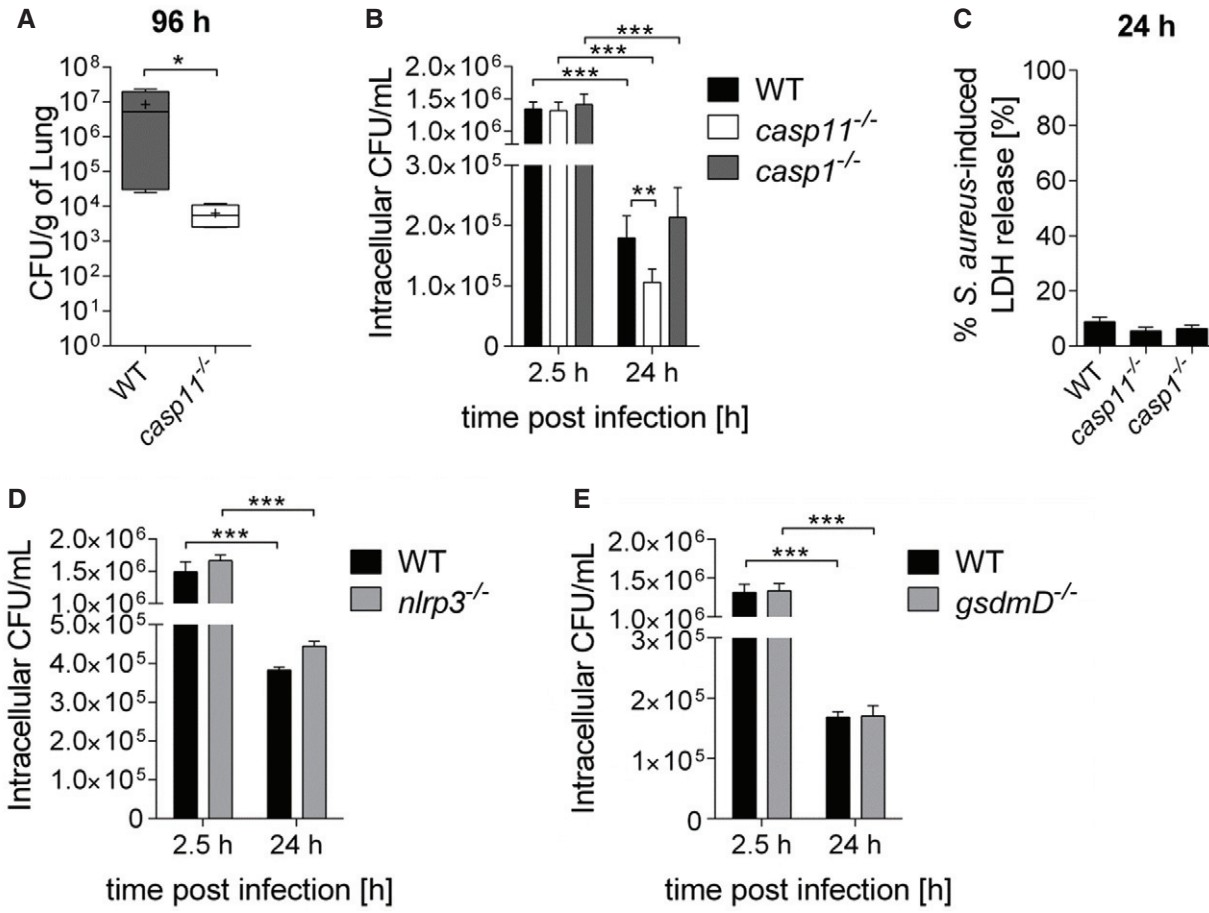

**Figure 3. CASP11 deficiency promotes MRSA clearance *in vitro* and *in vivo*.**

A  *In vivo* CFUs from lungs of WT and *casp11*[−/−] mice intratracheally infected with MRSA (2.5 × 10[8] CFUs/mouse) at 96 h post-infection (*n* = 4 biological replicates). Boxplot with whiskers from minimum to maximum. Horizontal bands represent the median, and "+" represents the mean. Statistical analysis was performed using two-tailed Student's *t*-test. **P* ≤ 0.05.

B  Intracellular survival of MRSA in WT, *casp11*[−/−], and *casp1*[−/−] BMDMs (MOI 5:1). Data represent mean ± SEM (*n* = 12 biological replicates). Statistical analysis was performed using a linear mixed effects model. ***P* ≤ 0.01, ****P* ≤ 0.001.

C  MRSA-induced LDH release in supernatants from WT, *casp11*[−/−], and *casp1*[−/−] BMDMs (MOI 5:1) at 24 h post-infection. Data represent mean ± SEM (*n* = 6 biological replicates). Statistical analysis was performed using one-way ANOVA.

D  Intracellular survival of MRSA in WT and *nlrp3*[−/−] BMDMs (MOI 5:1). Data represent mean ± SEM (*n* = 3 biological replicates). Statistical analysis was performed using a linear mixed effects model. ****P* ≤ 0.001.

E  Intracellular survival of MRSA in WT and *gsdmd*[−/−] BMDMs (MOI 5:1). Data represent mean ± SEM (*n* = 6 biological replicates). Statistical analysis was performed using a linear mixed effects model. ****P* ≤ 0.001.

superoxide levels in WT macrophages (Fig 5A). Macrophages lacking CASP11 exhibited higher MitoSOX fluorescence in response to MRSA than corresponding WT cells. The addition of Ant A led to a marked increase in MitoSOX fluorescence in both non-infected and MRSA-infected cells. These results suggest that mtROS production was increased in the absence of CASP11. To determine whether increased mtROS production triggered by Ant A further improves bacterial clearance of intracellular MRSA, we treated MRSA-infected WT and *casp11*[−/−] macrophages with Ant A for 24 h and enumerated intracellular bacterial numbers. In contrast to WT cells, wherein no effect could be detected, Ant A treatment significantly reduced the burden of intracellular MRSA in *casp11*[−/−] macrophages (Fig 5B). LDH release was significantly increased in response to Ant A, yet no difference in Ant A-induced cytotoxicity

could be found between WT and *casp11*[−/−] macrophages (Fig EV2C). There was also no increased extracellular bacterial growth in the presence of Ant A, indicating that gentamicin exclusion was still effective (Fig EV2D). In addition, treatment with the mtROS scavenger MitoQ elevated bacterial numbers in *casp11*[−/−] but not WT macrophages (Fig EV2E). This suggests that enhanced proximity to mitochondria contributes to MRSA elimination in response to Ant A-induced mitochondrial superoxide generation in *casp11*[−/−] macrophages.

Since mtROS production has been demonstrated to activate the NLRP3 inflammasome [32,33] and CASP1 activity was reported to control phagosomal acidification in response to Gram-positive bacteria [34], we subsequently examined whether Ant A-induced ROS production enhanced CASP1 activation, which might contribute

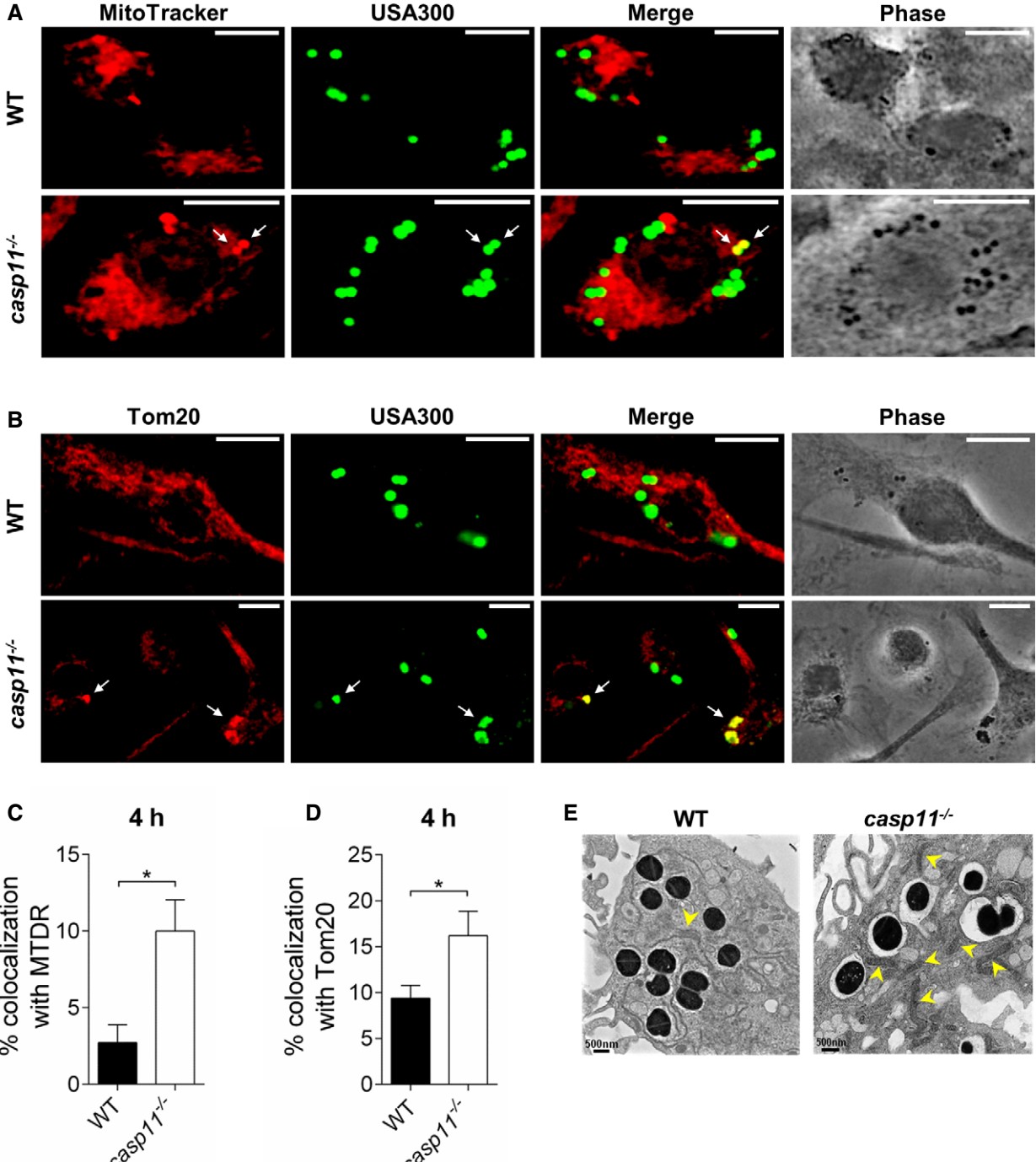

**Figure 4. Association of the MRSA-containing vacuole with mitochondria is increased in *casp11*<sup>−/−</sup> macrophages.**

A   MitoTracker Deep Red immunofluorescence assay of MRSA-infected WT and *casp11*<sup>−/−</sup> BMDMs at 4 h post-infection. White arrows indicate co-localization of MRSA with MitoTracker.

B   Tom20 immunofluorescence assay of MRSA-infected WT and *casp11*<sup>−/−</sup> BMDMs at 4 h post-infection. White arrows indicate co-localization of MRSA with Tom20.

C   Quantification of MRSA co-localized with MitoTracker Deep Red. Data represent mean ± SEM (*n* = 3 biological replicates). Statistical analysis was performed using two-tailed Student's *t*-test. *P ≤ 0.05.

D   Quantification of MRSA co-localized with Tom20. Data represent mean ± SEM (*n* = 3 biological replicates). Statistical analysis was performed using two-tailed Student's *t*-test. *P ≤ 0.05.

E   Qualitative transmission electron microscopy images of MRSA-infected WT and *casp11*<sup>−/−</sup> macrophages at 4 h post-infection. Yellow arrows indicate mitochondria.

Data information: Scale bars, 10 μm.

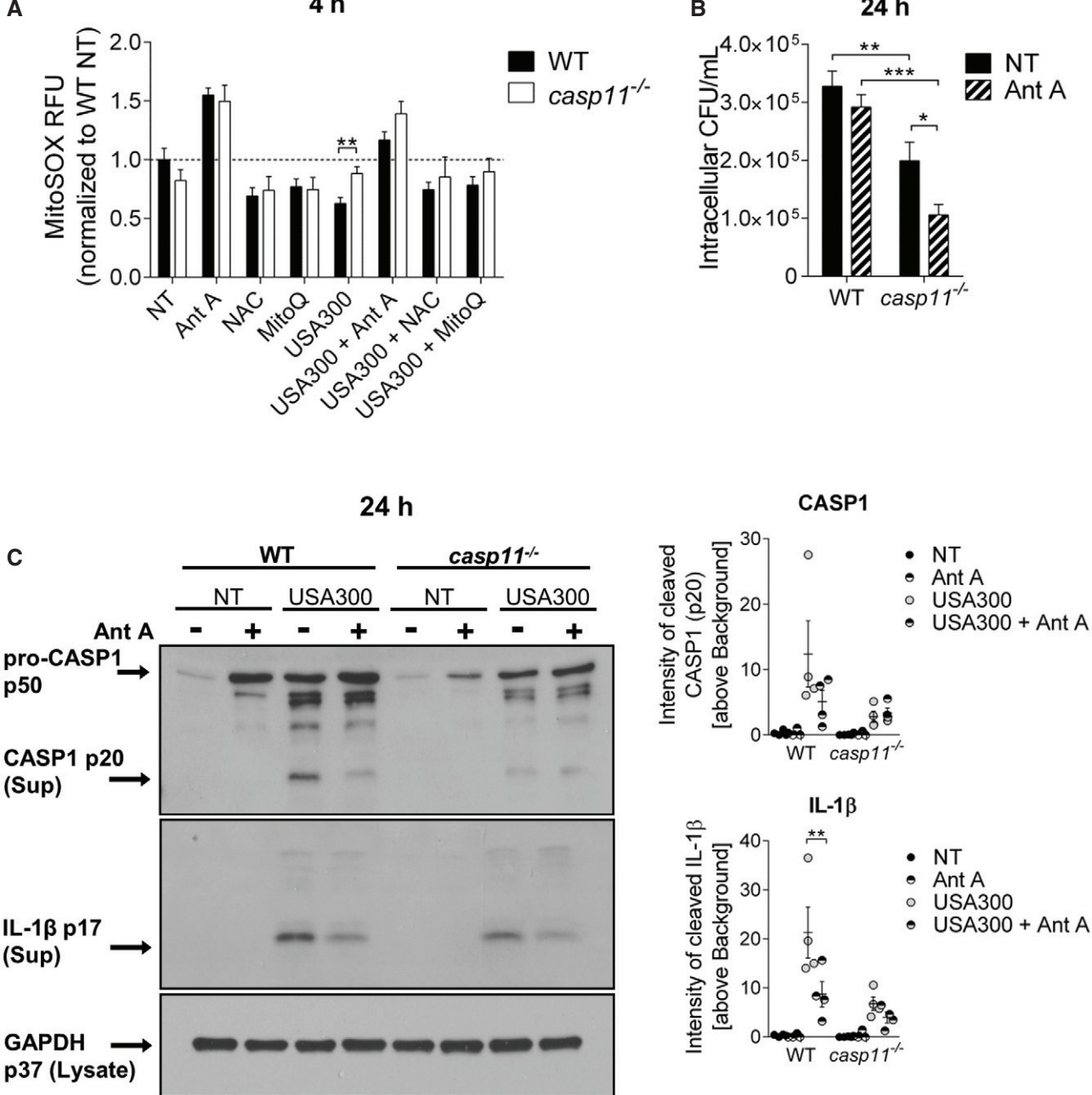

**Figure 5.  mtROS contributes to MRSA clearance in *casp11*⁻/⁻ macrophages.**

A   MitoSOX assay in WT and *casp11*⁻/⁻ BMDMs (MOI 5:1) at 4 h post-infection. Data represent mean ± SEM (*n* = 15 biological replicates). Statistical analysis was performed using multiple *t*-tests. **$P \leq 0.01$, NT = no treatment.

B   Intracellular CFU of MRSA (MOI 5:1) in WT and *casp11*⁻/⁻ BMDMs treated with Ant A at 24 h post-infection. Data represent mean ± SEM (*n* = 6 biological replicates). Statistical analysis was performed using two-way ANOVA. *$P \leq 0.05$, **$P \leq 0.01$, ***$P \leq 0.001$, NT = no treatment.

C   Immunoblot analysis of cleaved CASP1 and IL-1β in supernatants from WT and *casp11*⁻/⁻ BMDMs treated with Ant A at 24 h post-infection (MOI 20:1). Densitometry analysis represents mean ± SEM (*n* = 4 biological replicates). Statistical analysis was performed using two-way ANOVA. **$P \leq 0.01$, NT = no treatment.

to the reduced bacterial loads observed in *casp11*⁻/⁻ macrophages. As shown in Figs 5C and EV3A, Ant A treatment did not increase either CASP1 or IL-1β cleavage in WT and *casp11*⁻/⁻ macrophages, indicating that ROS-mediated killing of intracellular MRSA is independent of CASP1. In accordance with previous studies, which

demonstrated increased IL-8 production in response to oxidative stress [35,36], we found significantly higher amounts of CXCL1/KC produced by WT macrophages when treated with Ant A (Fig EV3A). No difference in the production of IL-1α or TNF was observed in response to Ant A (Fig EV3A). Furthermore, the inhibition of

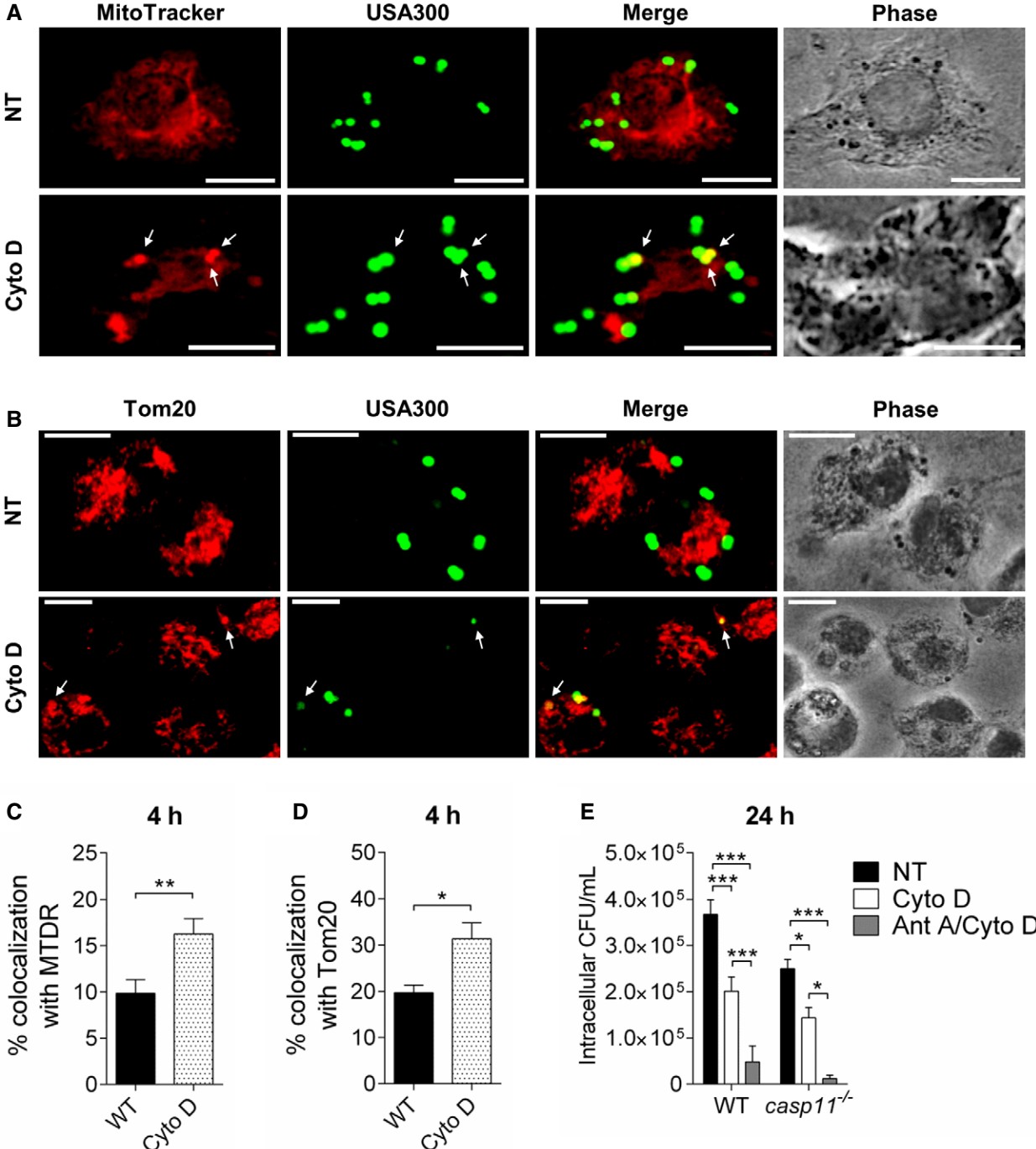

**Figure 6. Cyto D improves ROS-induced MRSA clearance in WT macrophages.**

A  MitoTracker Deep Red immunofluorescence assay of MRSA-infected WT BMDMs treated with Cyto D at 4 h post-infection. White arrows indicate co-localization of MRSA with MitoTracker.

B  Tom20 immunofluorescence assay of MRSA-infected WT BMDMs treated with Cyto D at 4 h post-infection. White arrows indicate co-localization of MRSA with Tom20.

C  Quantification of MRSA co-localized with MitoTracker Deep Red. Data represent mean ± SEM ($n$ = 4 biological replicates). Statistical analysis was performed using two-tailed Student's $t$-test. **$P$ ≤ 0.01.

D  Quantification of MRSA co-localized with Tom20. Data represent mean ± SEM ($n$ = 3 biological replicates). Statistical analysis was performed using two-tailed Student's $t$-test. *$P$ ≤ 0.05.

E  Intracellular CFU of MRSA in WT and *casp11*[−/−] BMDMs treated with Cyto D or Cyto D + Ant A at 24 h post-infection. Data represent mean ± SEM ($n$ = 6 biological replicates). Statistical analysis was performed using two-way ANOVA. *$P$ ≤ 0.05, ***$P$ ≤ 0.001, Data information: Scale bars, 10 μm.

lysosomal acidification using bafilomycin A1 (Baf A1) led to a marked decrease in bacterial loads in WT as well as $casp11^{-/-}$ macrophages (Fig EV3B), indicating that the acidification of bacteria-containing phagosomes is needed to allow MRSA survival in macrophages. Overall, these data suggest that bacterial co-localization with mitochondria and mtROS contribute to improved clearance of MRSA from $casp11^{-/-}$ macrophages, but it remains unclear why mitochondria show increased association with phagosomes in $casp11^{-/-}$ cells.

### Inhibition of the actin cytoskeleton prevents the dissociation of MRSA-containing phagosomes from mitochondria and leads to bacterial killing

CASP11 has been shown to mediate cell migration [37] and to modulate actin dynamics in response to bacterial infection [8,9]. Since a higher percentage of MRSA is associated with mitochondria in $casp11^{-/-}$ macrophages, we hypothesize that the inhibition of actin polymerization would facilitate co-localization of MRSA-containing phagosomes with mitochondria in WT cells. We, therefore, treated WT macrophages with Cyto D subsequent to MRSA infection and examined them by either employing the fluorescent MitoTracker Deep Red dye or staining with an anti-Tom20 antibody. In response to Cyto D, we found more bacteria co-localized with MitoTracker or Tom20 compared to non-treated cells (Fig 6A–D). Consistent with these findings, we also observed decreased bacterial loads at 24 h post-infection in both Cyto D-treated WT and $casp11^{-/-}$ macrophages (Fig 6E). Cyto D had no direct bactericidal activity against MRSA (Fig EV2A). Furthermore, the combination of Cyto D with Ant A led to an additive effect and a greater reduction in bacterial numbers as compared to Cyto D treatment alone (Fig 6E). While LDH release was comparable between non-treated and Cyto D-treated WT and $casp11^{-/-}$ macrophages infected with MRSA (Fig EV2C), there was no difference in extracellular bacterial loads (Fig EV2D). Similar to Ant A, testing for CASP1 and IL-1β cleavage revealed no elevated inflammasome activity in response to Cyto D (Figs 7A and EV3C). Additionally, to test whether F-actin assembles around the MRSA-containing phagosome, we stained WT and $casp11^{-/-}$ macrophages with phalloidin. Interestingly, co-localization of MRSA with phalloidin was mildly decreased in $casp11^{-/-}$ macrophages (Fig 7B and C). Together, these results suggest that impaired actin dynamics prevent the dissociation of mitochondria from internalized MRSA, thereby facilitating mtROS-driven bacterial killing.

### mtROS production but not impaired autophagy contributes to MRSA clearance in $casp11^{-/-}$ macrophages

Autophagy/macroautophagy is a host cell defense mechanism that has been implicated in the degradation of various intracellular

pathogens within double-membrane structures called autophagosomes. Bacteria, including *S. aureus*, have developed strategies to escape from or even subvert the autophagic machinery [38]. Since autophagosomes have been demonstrated to support *S. aureus* survival in HeLa cells [38], we examined the impact of pharmacological stimulation or inhibition of autophagy on intracellular MRSA survival in WT macrophages using rapamycin (mTOR inhibitor) or wortmannin (PI3K inhibitor), respectively. To circumvent the inhibitory effect of wortmannin on bacterial host cell entry, the treatment was given after phagocytosis of MRSA was completed. Consistent with previous findings, bacterial loads at 24 h post-infection were mildly but not significantly increased in response to the autophagy-stimulating drug rapamycin (Fig 8A). In contrast, wortmannin significantly reduced the intracellular burden of MRSA, suggesting that the suppression of autophagy by wortmannin [39,40] among other effects facilitates MRSA clearance from murine macrophages. Rapamycin had minor bactericidal activity, whereas no effect could be observed for wortmannin (Fig EV2A). Notably, in addition to its function as inhibitor of mitochondrial respiration, Ant A has been reported to negatively affect the autophagy pathway [41]. Since we found reduced bacterial loads of MRSA in Ant A-treated $casp11^{-/-}$ macrophages, we questioned whether compromised autophagy in response to Ant A might contribute to improved bacterial clearance in the absence of CASP11. To elucidate the role of Ant A in the autophagic process during MRSA infection, we compared the levels of the autophagosomal marker protein LC3-II in WT and $casp11^{-/-}$ macrophages either untreated or treated with Ant A via Western blot analysis. In accordance with previous findings [41], Ant A reduced the levels of LC3-II in both non-infected and MRSA-infected WT and $casp11^{-/-}$ macrophages (Fig 8B). However, Ant A reduced bacterial numbers only in $casp11^{-/-}$ macrophages. Since $casp11^{-/-}$ macrophages exhibit increased co-localization of MRSA with mitochondria (Fig 4A–D), this suggests that the stimulation of mtROS production by Ant A is the major cause for effective MRSA clearance, not the inhibition of autophagy.

In contrast, disruption of the mitochondrial membrane potential (MMP) through uncoupling agents, such as trifluoromethoxy carbonylcyanide phenylhydrazone (FCCP), has been demonstrated to induce both autophagy and selective mitophagy [42,43]. Correspondingly, FCCP treatment led to a significant increase in LC3-II formation in WT and $casp11^{-/-}$ macrophages (Fig 8C). Yet, an increased bacterial burden due to the stimulation of autophagy via FCCP could be observed only in WT macrophages where MRSA-containing phagosomes do not co-localize with mitochondria (Fig 8D). Since FCCP had a direct bactericidal effect against MRSA (Fig EV2A), FCCP treatment was given post-MRSA infection. Overall, while inhibiting autophagy can further augment the clearance of intracellular MRSA, we conclude that improved bactericidal activity of $casp11^{-/-}$ macrophages against MRSA is mainly attributed to

---

**Figure 7. Cyto D improves MRSA clearance in WT macrophages independent of CASP1.**

A  Immunoblot analysis of cleaved CASP1 and IL-1β in supernatants from WT and $casp11^{-/-}$ BMDMs treated with Cyto D at 24 h post-infection. Densitometry analysis represents mean ± SEM (*n* = 4 biological replicates). Statistical analysis was performed using two-way ANOVA. NT = no treatment.
B  Rhodamine phalloidin immunofluorescence assay of MRSA-infected WT and $casp11^{-/-}$ BMDMs at 4 h post-infection.
C  Quantification of MRSA co-localized with rhodamine phalloidin. Data represent mean ± SEM (*n* = 3 biological replicates). Statistical analysis was performed using two-tailed Student's *t*-test. *$P \leq 0.05$.

Data information: Scale bars, 10 μm.

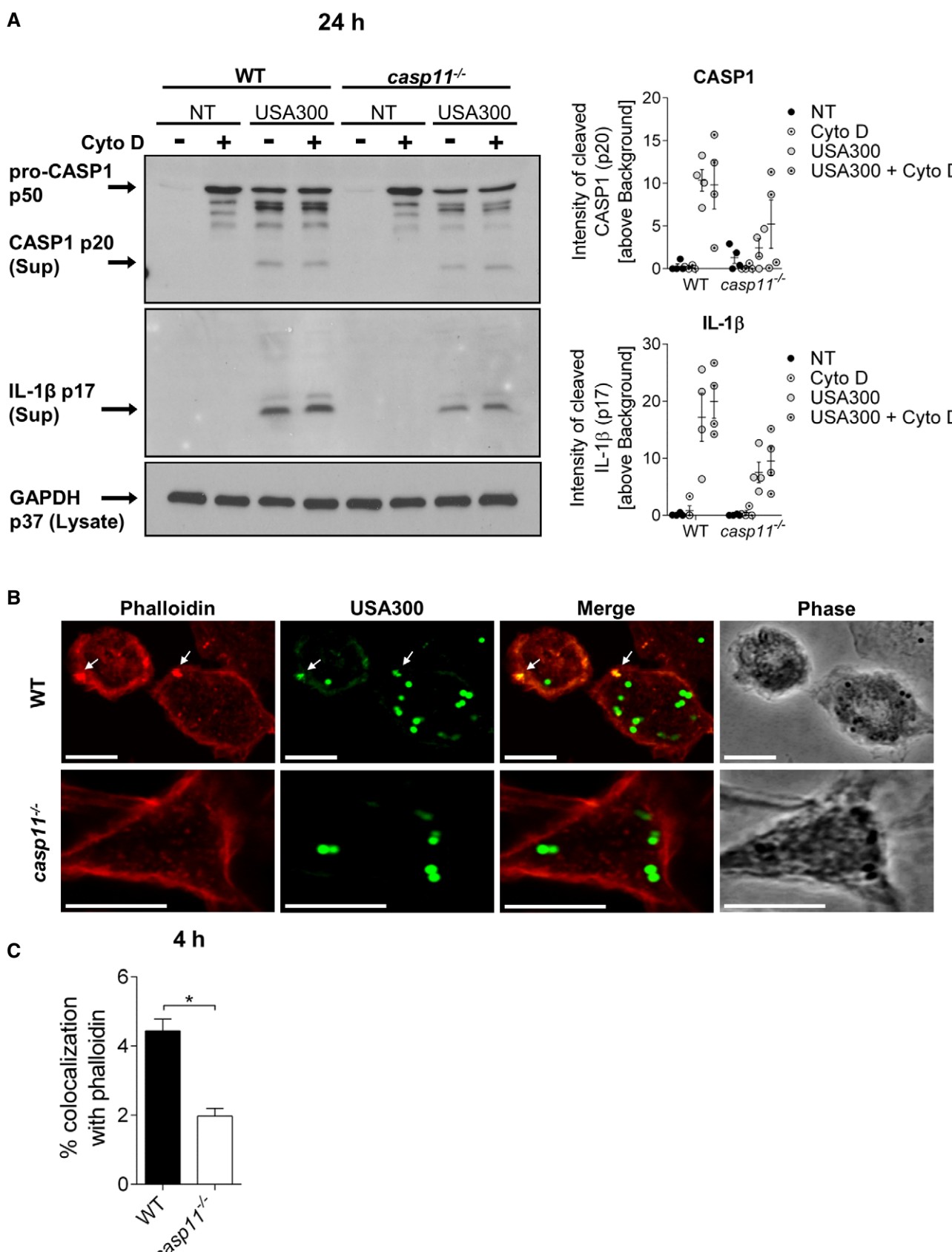

**Figure 7.**

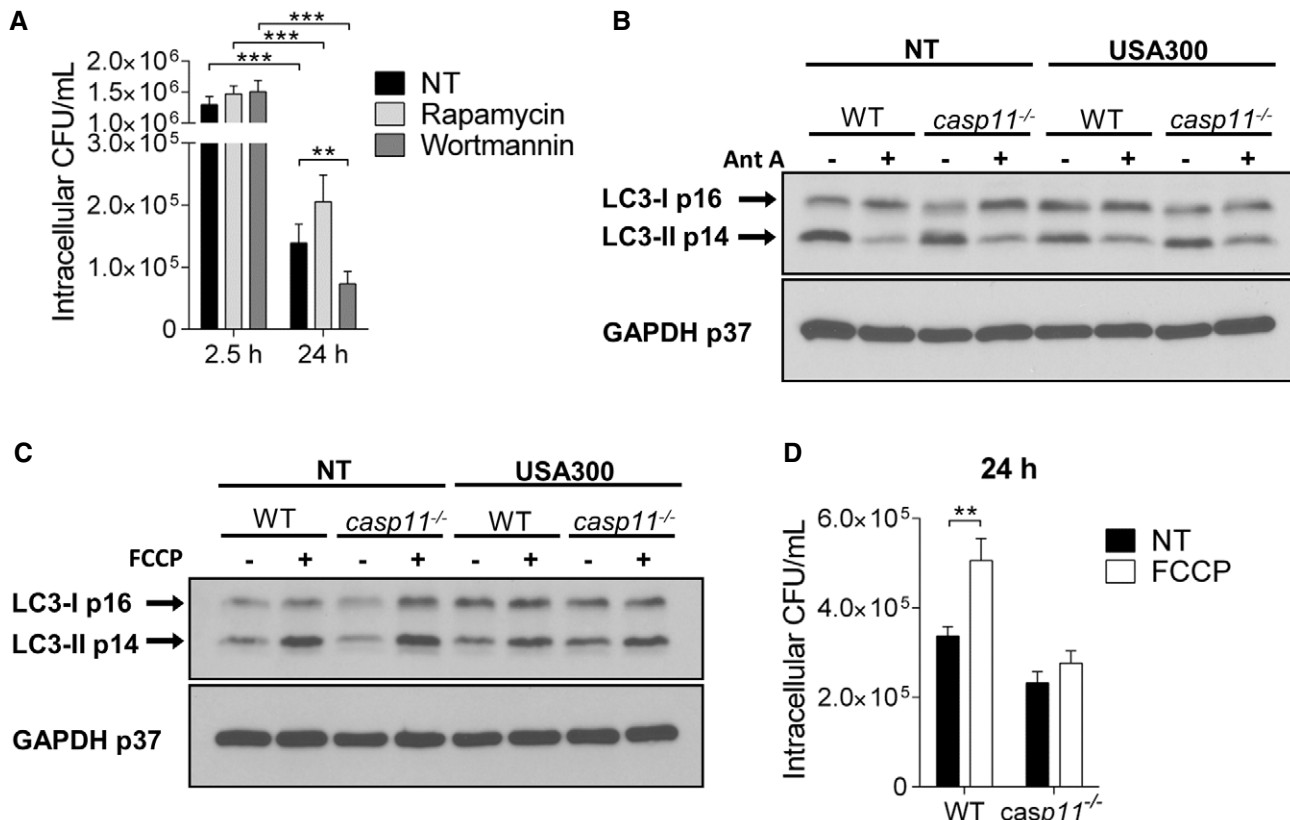

**Figure 8. Compromised autophagy contributes to the clearance of intracellular MRSA.**

A   Intracellular survival of MRSA in WT BMDMs treated with rapamycin or wortmannin. Data represent mean ± SEM (*n* = 5 biological replicates). Statistical analysis was performed using a linear mixed effects model. **$P \leq 0.01$, ***$P \leq 0.001$, NT = no treatment.

B   Representative immunoblot of LC3-II in cell lysates from WT and *casp11*[−/−] BMDMs treated with Ant A at 24 h post-infection (*n* = 3 biological replicates).

C   Representative immunoblot of LC3-II in cell lysates from WT and *casp11*[−/−] BMDMs treated with FCCP at 24 h post-infection (*n* = 3 biological replicates).

D   Intracellular survival of MRSA in WT and *casp11*[−/−] BMDMs treated with FCCP. Data represent mean ± SEM (*n* = 4 biological replicates). Statistical analysis was performed using two-way ANOVA. **$P \leq 0.01$, NT = no treatment.

localized mitochondria-derived ROS in close proximity to bacteria-containing phagosomes.

## Discussion

*Staphylococcus aureus* exhibits numerous virulence factors that have been reported to induce multiple inflammasomes. Specifically, the activation of the NLRP3 inflammasome has been shown in response to peptidoglycan or pore-forming toxins such as Hla, leukocidin AB (Leu AB), and Panton–Valentine leukocidin (PVL), leading to cleavage of CASP1, IL-1β/IL-18 secretion, and pyroptosis [21,22,44–46]. In addition, AIM2 and ASC have a protective role during acute central nervous infection with *S. aureus* [47]. Although CASP11 is well known to mediate non-canonical NLRP3 inflammasome signaling during Gram-negative bacterial infection, a possible contribution of CASP11 in *S. aureus*-induced inflammasome activation was not previously discussed. It is noteworthy that experimental systems investigating the role of NLRP3 during *S. aureus* infection often involved priming steps with either LPS or LTA [20,44,45], which are known to induce CASP11 expression through TLR4 or TLR2

signaling [6,48]. Furthermore, *L. monocytogenes* and *S. aureus* infection, as well as transfection of LTA, were recently demonstrated to activate CASP11 in murine macrophages [15]. Therefore, it is plausible to conclude that CASP11 is needed for CASP1 processing and downstream events in response to *S. aureus*. Although in our study we did not observe cleavage of CASP11 in macrophages infected with MRSA, which might be attributed to differences between *S. aureus* strains, we found a significant reduction of cleaved CASP1 and IL-1β in the absence of CASP11. Furthermore, we report a CASP11-dependent release of activated CASP7 in cell culture supernatants. Cleavage of CASP7 during *S. aureus* infection has been described for keratinocytes and THP-1 macrophages [49,50], and in response to Hla [51]. In the context of *L. monocytogenes* infection, CASP7 prevents plasma membrane damage caused by the pore-forming toxin listeriolysin O (LLO) [51]. Inflammasome-dependent activation of CASP7 results in cleavage and inactivation of Poly (ADP-ribose) polymerase 1 (PARP1) [52,53], thereby allowing pro-inflammatory NF-κB target gene expression. *L. pneumophila*-induced CASP7 activation facilitates phagosome–lysosome fusion [26]. However, the outcome of CASP7 activity during *S. aureus* infection remains uncharacterized.

Since previous reports suggested a potential role of the necroptosis effector protein MLKL for IL-1β secretion in human THP-1 macrophages in response to MRSA [24], we analyzed processing of CASP1, IL-1β, and CASP7 in MLKL-deficient BMDMs. However, we did not observe any difference in the maturation or secretion of these proteins in murine macrophages lacking MLKL in response to MRSA. In contrast, increased bacterial burdens, CASP1 activation, and IL-1β release were reported in skin biopsies from $mlkl^{-/-}$ mice upon subcutaneous MRSA infection [50]. Therefore, it is possible that inflammasome signaling pathways activated in response to MRSA differ between cell types.

GSDMD cleaved by CASP11 or CASP1 promotes pyroptosis and IL-1β secretion [11,54,55]. Specifically, oligomerization of the N-terminal GSDMD fragment leads to the formation of a plasma membrane pore allowing the release of cytosolic content [56,57]. Consistent with these findings, we demonstrate lower levels of cleaved CASP1, IL-1β, and CASP7 in cell culture supernatants of MRSA-infected $gsdmd^{-/-}$ macrophages compared to corresponding WT cells. Yet, in comparison to $casp11^{-/-}$ and $casp1^{-/-}$ macrophages, we found mildly increased amounts of IL-1β and CASP7 cleavage products in $gsdmd^{-/-}$ macrophage supernatants. Although the immediate release of IL-1β in response to inflammasome activation was shown to require GSDMD, there is also evidence for a GSDMD-independent mechanism of IL-1β secretion through phosphatidylinositol 4,5-bisphosphate (PIP2) enriched plasma membrane domains [58]. Since MRSA-induced processing of IL-1β and CASP7 still occurs in macrophages lacking GSDMD due to active CASP1, our results indicate that the secretion of their mature products in $gsdmd^{-/-}$ macrophages is indeed rather delayed but not prevented.

Intracellular survival of *S. aureus* has been demonstrated for various cell types, including macrophages, neutrophils, and endothelial cells [29,59–61]. Notably, multiple studies have reported that *S. aureus* exploits autophagy to either replicate within autophagosomes or scavenge nutrients [38,62–64]. Accordingly, we found increased survival of MRSA in response to the autophagy-stimulating drugs rapamycin and FCCP, whereas the inhibition of autophagy via wortmannin or Ant A promoted bacterial clearance. The persistence of MRSA within host cells provides the bacterium with protection against antibiotics [61]. When the antibiotic pressure is removed, MRSA escapes from host cells followed by extracellular replication and infection of neighboring cells. Hence, it is necessary to identify mechanisms that allow efficient clearance of intracellular MRSA. Recently, mitochondria-derived ROS were shown to exert antimicrobial effects on intracellular MRSA [16,31]. There is emerging evidence that elevated production of mtROS drives the bactericidal activity of macrophages against intracellular bacterial pathogens such as *S.* Typhimurium, *L. monocytogenes*, *M. tuberculosis,* and *E. coli* [17,19,65,66]. Interestingly, Hla-induced NLRP3 inflammasome activation promotes active dissociation of mitochondria from intracellular bacteria and thus contravenes mtROS-driven eradication of MRSA [16]. The recruitment and association of NLRP3 to and with mitochondria through direct interaction with mitochondrial cardiolipin [67,68] indicates that mitochondria act as scaffolds to facilitate NLRP3 inflammasome activation. Although no difference in the bacterial burden of *S. aureus* between WT and $nlrp3^{-/-}$ mice was found in an earlier report [23], the inhibition of NLRP3 in human monocytes improves killing of intracellular MRSA due to increased bacterial association with mitochondria [16]. Consequently, in the absence of Hla and NLRP3 activation, co-localization of MRSA with mitochondria is more pronounced [16]. Here, we show that the disengagement of MRSA-containing phagosomes from mitochondria is also reduced in macrophages deficient of CASP11, allowing mtROS to contribute to bacterial clearance. Interestingly, we observed a decline in mitochondrial superoxide production in response to MRSA infection that was dependent on CASP11 expression in macrophages. In addition, elevated amounts of superoxide induced by Ant A exert bactericidal effects only in the absence of CASP11 where bacteria are found in the proximity of mitochondria. Notably, MRSA has been demonstrated to trigger mitochondria-derived vesicle (MDV) formation in infected macrophages, leading to the delivery of the hydrogen peroxide-generating enzyme superoxide dismutase-2 (SOD2) to bacteria-containing phagosomes [31]. Since $casp11^{-/-}$ macrophages control bacterial loads more effectively in response to Ant A than corresponding WT cells, it is plausible to hypothesize that CASP11 might interfere with either MDV production or SOD2 delivery to the MRSA-containing phagosome.

*Staphylococcus aureus* carries multiple genes encoding ROS-detoxifying enzymes, including catalase and superoxide dismutase, to minimize the effects of oxidative stress [3,69]. Our results suggest that *S. aureus* defense mechanisms are less effective in macrophages lacking CASP11. The significant role of mtROS for MRSA eradication is further confirmed by the fact that the treatment of WT macrophages with Cyto D prevents MRSA dissociation from mitochondria, leading to improved bacterial clearance. Both actin filaments and microtubules have been implicated in the movement of mitochondria [70,71]. Considering that multiple reports describe actin dynamics to be defective in CASP11-deficient cells [8,13,37], it is conceivable that CASP11 promotes the separation of MRSA-containing phagosomes from mitochondria through the regulation of actin filaments.

In addition to mtROS, CASP1 was demonstrated to promote the killing of Hla-deficient *S. aureus* by modulating phagosomal acidification [16,34]. However, others have shown that *S. aureus* resides and replicates in acidic compartments [72,73], indicating that acidity does not reduce MRSA survival within macrophages. In our study, cleavage of CASP1 and subsequent IL-1β release did not increase either in response to Ant A or Cyto D. In addition, there was no difference in the bacterial burden between WT and $casp1^{-/-}$ macrophages, and the inhibition of acidification using Baf A1 significantly reduced MRSA replication, indicating that acidification is required for MRSA persistence. We therefore conclude that CASP11 promotes MRSA persistence since it is required for the active dissociation of mitochondria away from the vicinity of MRSA-containing vacuoles. Contrary to Gram-positive bacteria, CASP11 promotes the clearance of Gram-negative bacteria by promoting phagosome–lysosome fusion [8,9,13]. Our data provide a new role for CASP11 in the pathogenesis of Gram-positive bacteria.

## Materials and Methods

### Bacterial strains

*Staphylococcus aureus* strain USA300 was used in this study. USA300 complemented with a plasmid for green fluorescent protein

(GFP) (BEI resources) was used in immunofluorescence experiments. All bacterial strains were grown in tryptic soy broth (TSB) at 37°C and 200 rpm overnight, and sub-cultured the next day before use in experiments.

## Mice

C57BL/6 WT mice were obtained from the Jackson Laboratory (Bar Harbor, ME, USA). $Casp11^{-/-}$ mice were generously given by Dr. Yuan at Harvard Medical School, Boston, MA, USA [74]. $Casp-1^{-/-/Casp-11Tg}$ mice were kindly provided by Dr. Vishva Dixit at Genentech, San Francisco, CA, USA. $Gsdmd^{-/-}$, $mlkl^{-/-}$, and $nlrp3^{-/-}$ mice were a gift from Dr. Thirumala-Devi Kanneganti at St. Jude Children's Research Hospital, Memphis, TN, USA. All mice were housed in a pathogen-free facility, and experiments were conducted with approval from the Animal Care and Use Committee at the Ohio State University (Columbus, OH, USA).

## In vivo infection

For intratracheal infection, mice were anesthetized with Isoflurane and inoculated with 100 μl of PBS containing $2.5 \times 10^8$ CFUs of MRSA USA300. To determine the bacterial load in organs, mice were sacrificed at 96 h post-infection to collect lungs, for homogenization in PBS as previously described [9,13,75]. Data are presented as CFUs per gram of organ tissue.

## Cell culture

For the generation of primary bone marrow-derived macrophages (BMDMs) from mice, tibias and femurs were flushed with IMDM media (Thermo Fisher Scientific, 12440053) supplemented with 10% of heat inactivated fetal bovine serum (FBS, Thermo Fisher Scientific, 16000044), 50% L cell-conditioned media, 0.6× MEM Non-Essential Amino Acids (Thermo Fisher Scientific, 11140050), and 0.1% penicillin and streptomycin (Thermo Fisher Scientific, 15140122). Cells were cultivated at least 6 days at 37°C in a humidified atmosphere containing 5% $CO_2$ as previously described [8,9,13,76]. THP-1 cells either expressing Cas9 or co-expressing $CASP4^{-/-}$, $CASP5^{-/-}$, $CASP4/5^{-/-}$, and $CASP1^{-/-}$ were maintained in RPMI 1640 media (Thermo Fisher Scientific, 22400105) supplemented with 10 % FBS and 0.1% penicillin and streptomycin. To induce the CRISPR system, cells were incubated with 1 μg/ml of doxycycline hyclate (Sigma, D9891) for 72 h and the knockout status was verified by immunoblotting [27]. THP-1 monocytes were differentiated into macrophages by 48-h incubation with 25 nM phorbol 12-myristate 13-acetate (PMA, Sigma-Aldrich, P8139) followed by 24-h incubation in RPMI medium plus 10% FBS.

## In vitro infection of primary macrophages

Prior to infection, macrophages were cultivated in IMDM or RPMI media supplemented with 10% FBS. In vitro infections were performed with MOI 5:1 or MOI 20:1 as indicated, including centrifugation for 5 min at 200× g to synchronize the infection. For gentamicin protection assays and protein determination via Western blot analysis in cell culture lysates or supernatants, macrophages were infected for 1 h followed by two gentamicin treatments of cells

(1 h followed by 30 min of incubation, respectively, with 100 and 40 μg/ml of gentamicin) to eliminate extracellular bacteria. Macrophages were then incubated with 10 μg/ml gentamicin until the end of the experiment in the presence or absence of Ant A (5 μM), Cyto D (2 μM), FCCP (5 μM), rapamycin (5 μg/ml), wortmannin (200 nM), NAC (3 mM), MitoQ (1 μM), or Baf A1 (100 nM) as indicated. For the determination of intracellular bacterial loads, macrophages were lysed at indicated time points using 0.1% Triton X-100 (Fisher Scientific, BP151) in PBS, and CFUs of MRSA were determined in serial dilutions of the lysates using TSB agar plates incubated for 24 h.

## Confocal microscopy

Macrophages were cultured on glass coverslips in 24-well plates and fixed with 4% paraformaldehyde for 30 min at indicated time points. Mitochondria of infected macrophages were stained for 15 min with 250 nM of MitoTracker™ Deep Red (Molecular Probes, M22426). MitoTracker™ Deep Red staining was pseudocolored in red. For permeabilization, cells were treated with 0.3% Triton X-100 for 20 min followed by blocking with 5% goat serum (Thermo Fisher Scientific, 16210064) in PBS. Tom20 (Sigma-Aldrich, WH0009804M1) was visualized using goat anti-mouse IgG secondary antibody conjugated to Alexa Fluor® 594 (Molecular Probes, A-11032). Nuclei were counterstained with 1 μg/ml of Hoechst solution (Molecular Probes, 62249) for 15 min. Rhodamine phalloidin (Molecular Probes, R415) was used to stain F-actin filaments for 20 min. Images were captured using a laser scanning confocal fluorescence microscope with a 60× objective (Olympus Fluoview FV10i) as previously described [8,76].

## ELISA

Cytokines in cell culture supernatants were measured by R&D Systems DuoSet ELISA Development Systems (murine IL-1α, DY400, murine IL-1β, DY401, murine CXCL1/KC, DY453, murine TNF-alpha, DY410, human IL-1α, DY200, human IL-1β, DY201, human IL-8, DY208, human TNF-alpha, DY210) according to the manufacturer's instructions.

## Immunoblotting

Protein extraction from macrophages was performed using TRIzol reagent according to the manufacturer's instructions. Briefly, after phase separation using chloroform, 100% ethanol was added to the interphase/phenol–chloroform layer to precipitate genomic DNA. Subsequently, the phenol–ethanol supernatant was mixed with isopropanol to isolate proteins. The Bradford method was used to determine protein concentrations in cell lysates. Equal amounts of protein or supernatants were separated by 13.5% SDS–PAGE and transferred to a polyvinylidene fluoride (PVDF) membrane. Membranes were incubated overnight with antibodies against CASP11 (Cell Signaling, 14340), murine CASP1 (Adipogen, AG-20B-0042-C100), human cleaved CASP1 (Cell Signaling Technology, 4199), human CASP1 (Cell Signaling Technology, 2225), cleaved CASP7 (Cell Signaling Technology, 9491), human CASP4 (MBL, M0293), murine IL-1β (R&D Systems, AF-401-NA), and GAPDH (Cell Signaling Technology, 2118). Corresponding secondary

antibodies conjugated with horseradish peroxidase and in combination with enhanced chemiluminescence reagent were used to visualize protein bands. Densitometry analyses were performed by normalizing target protein bands to their respective loading control (GAPDH) using ImageJ software as previously described [8,9].

### LDH assay

LDH release from macrophages infected with MRSA was measured using the CytoTox-ONE Homogeneous Membrane Integrity Assay (Promega, G7891) according to the manufacturer's instructions. MRSA-induced LDH release [%] = ((infected sample-low control)/ (high control-low control))*100.

### Mitochondrial ROS assay

To determine mitochondrial superoxide production, macrophages were incubated for 30 min with 2 μM MitoSOX dye (Thermo Fisher Scientific, M36008) diluted in cell imaging solution (10 mM Hepes, 1 mg/ml BSA, 1 mg/ml Glucose, 1 mM $MgCl_2$, 1.8 mM $CaCl_2$ in PBS) at 37°C in a humidified atmosphere containing 5% $CO_2$. Then, cells were washed with PBS and further incubated with cell imaging solution containing 10 μg/ml gentamicin. The fluorescence was read using a SpectraMax i3x microplate reader at 510/580 nm, and the cell number in each well was determined using a SpectraMax MiniMax 300 Imaging Cytometer. Fluorescence values were first normalized to the cell number in each well, and then, all conditions for each genotype were normalized to the WT non-treated control group.

### Transmission electron microscopy

Macrophages were cultured in 2-well Permanox Lab-Tek chamber slides (Nunc, 177429) and fixed with 2.5% glutaraldehyde (Ted Pella, 18426) in 0.1 M phosphate buffer, pH 7.4 (Fisher Scientific, S369 and S373) containing 0.1 M sucrose (Fisher Scientific, S2-500). Sample processing was performed by the Campus Microscopy & Imaging Facility at The Ohio State University as previously described [75]. Images were taken using a FEI Tecnai G2 Spirit transmission electron microscope plus AMT camera system.

### Statistical analysis

Data were analyzed using GraphPad Prism 6.0. All figures display mean and standard error of the mean (SEM) from at least three independent experiments as indicated in the figure legends. Comparisons between groups were conducted with either Student's *t*-test, one-way or two-way ANOVA, or linear mixed effects model (depending on the data structure) followed by Holm's adjustment for multiple comparisons as indicated. Adjusted $P < 0.05$ was considered statistically significant.

**Expanded View** for this article is available online.

### Acknowledgements
We thank Dr. Seth Masters, Walter and Eliza Hall Institute of Medical Research, Australia, for providing THP-1 cells either expressing Cas9 or co-expressing *CASP4$^{-/-}$*, *CASP5$^{-/-}$*, *CASP4/5$^{-/-}$*, and *CASP1$^{-/-}$*. Studies in the Amer laboratory are supported by NIAID R01 AI24121 and NHLBI R01 HL127651-01A1. KK was supported by Deutsche Forschungsgemeinschaft (DFG; German Research Foundation) and then by a Cystic Fibrosis Postdoctoral Research Fellowship. AB and SE are supported by funding from the Egyptian Bureau of Education.

### Author contributions
Conceptualization, AOA, VP, and KK; Formal Analysis, KK; Investigation, KK, KD, SE, AB, KH, RH, MNKA, BK, XZ, MAG, and VP; Resources, AOA; Writing-Original draft, KK; Writing-Review & Editing, KK, AOA, KD, SE, AB, KH, AAK, RH, XZ, MAG, and VP; Project Administration, AOA and KK; Supervision, AOA; Funding Acquisition, AOA.

### Conflict of interest
The authors declare that they have no conflict of interest.

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
