## [Review Process File · EMBO Reports]

Caspase-11 counteracts mitochondrial ROS-mediated clearance of *Staphylococcus aureus* in macrophages

Kathrin Krause, Kylene Daily, Shady Estfanous, Kaitlin Hamilton, Asmaa Badr, Arwa Abu Khweek, Rana Hegazi, Midhun N. K. Anne, Brett Klamer, Xiaoli Zhang, Mikhail A. Gavrilin, Vijay Pancholi, and Amal O. Amer

Review timeline:

Submission date:	17 March 2019
Editorial Decision:	16 April 2019
Revision received:	15 July 2019
Editorial Decision:	16 August 2019
Revision received:	16 September 2019
Accepted:	25 September 2019

Editor: Achim Breiling

Transaction Report:

1st Editorial Decision

16 April 2019

Thank you for the submission of your research manuscript to EMBO reports. We have now received reports from the three referees that were asked to evaluate your study, which can be found at the end of this email.

As you will see, all referees think the manuscript is of high interest, but requires major revisions to allow publication in EMBO reports. As the reports are below, I will not further detail them here. However, I feel that in particular points 1 and 2 of referee #1 and points 1, 3, 4, 5 and 7 of referee #2 need particular attention. We would not require further mechanistic insight (e.g. point 6 of referee #2 and minor point 1 of referee #1).

Given the constructive referee comments, we would like to invite you to revise your manuscript with the understanding that all referee concerns must be addressed in the revised manuscript and in a detailed point-by-point response. Acceptance of your manuscript will depend on a positive outcome of a second round of review. It is EMBO reports policy to allow a single round of revision only and acceptance or rejection of the manuscript will therefore depend on the completeness of your responses included in the next, final version of the manuscript.

Revised manuscripts should be submitted within three months of a request for revision; they will otherwise be treated as new submissions. Please contact me if a 3-months time frame is not sufficient so that we can discuss the revisions further.

Supplementary/additional data: The Expanded View format, which will be displayed in the main HTML of the paper in a collapsible format, has replaced the Supplementary information. You can submit up to 5 images as Expanded View. Please follow the nomenclature Figure EV1, Figure EV2

etc. The figure legend for these should be included in the main manuscript document file in a section called Expanded View Figure Legends after the main Figure Legends section. Additional Supplementary material should be supplied as a single pdf labeled Appendix. The Appendix includes a table of content on the first page, all figures and their legends. Please follow the nomenclature Appendix Figure Sx throughout the text and also label the figures according to this nomenclature.

For more details please refer to our guide to authors:
<http://embor.embopress.org/authorguide#manuscriptpreparation>

Please add a running title (not more than 40 characters including spaces) to the title page, and a conflict of interest statement below the author contributions.

Important: All materials and methods should be included in the main manuscript file.

See also our guide for figure preparation:
http://www.embopress.org/sites/default/files/EMBOPress_Figure_Guidelines_061115.pdf

Regarding data quantification and statistics, can you please specify, where applicable, the number "n" for how many independent experiments (biological replicates) were performed, the bars and error bars (e.g. SEM, SD) and the test used to calculate p-values in the respective figure legends. Please provide statistical testing where applicable. See:
<http://embor.embopress.org/authorguide#statisticalanalysis>

Please also follow our guidelines for the use of living organisms, and the respective reporting guidelines: <http://embor.embopress.org/authorguide#livingorganisms>

We now strongly encourage the publication of original source data with the aim of making primary data more accessible and transparent to the reader. The source data will be published in a separate source data file online along with the accepted manuscript and will be linked to the relevant figure. If you would like to use this opportunity, please submit the source data (for example scans of entire gels or blots, data points of graphs in an excel sheet, additional images, etc.) of your key experiments together with the revised manuscript. Please include size markers for scans of entire gels, label the scans with figure and panel number, and send one PDF file per figure.

- a complete author checklist, which you can download from our author guidelines (<http://embor.embopress.org/authorguide#revision>). Please insert page numbers in the checklist to indicate where the requested information can be found.
- a letter detailing your responses to the referee comments in Word format (.doc)
- a Microsoft Word file (.doc) of the revised manuscript text
- editable TIFF or EPS-formatted single figure files in high resolution (for main figures and EV figures)

Please also note that we now mandate that the corresponding authors provide an ORCID digital identifier that is linked to his/her EMBO reports account.

I look forward to seeing a revised version of your manuscript when it is ready. Please let me know if you have questions or comments regarding the revision.

REFeree REPORTS

Referee #1:

The manuscript by Krause et al. describes the role of Caspase11 in macrophage innate immunity against MRSA. The manuscript shows two major findings that occur independently of each other. First, Caspase11 expression is induced during MRSA infection and contributes to pro-inflammatory

cytokine production (IL-1, IL-1 and Cxcl1/ KC). Caspase11 deficient macrophages produce less pro-inflammatory cytokines when compared to WT macrophages. Second, the authors suggest that loss of Caspase11 increases macrophage bactericidal activity by enhancing mitochondrial association with MRSA-containing phagosomes. Mitochondrial association with phagosomes and enhancing macrophage bactericidal activity has been previously described for Gram-negative bacteria (West, et al, Nature 2011). This study evaluates a potential role for Caspase11 in controlling this process. The authors propose that Caspase11 facilitates MRSA evasion through macrophage killing with mROS.

The manuscript is clearly written and the authors used a robust genetic approach to assess the role of Caspase11 in pro-inflammatory cytokine production during MRSA infection. Although the role of Caspase11 in activation of the non-canonical inflammasome has been previously described within the context of other infection models, especially Gram-negative bacteria, this study establishes the importance of Caspase11 in non-canonical inflammasome activation during infection by MRSA, a pathogen of public health significance. The conclusions of this part of the study are well supported by the data. However, the second finding of the study, regarding the role of Caspase11 in mitochondria-mediated MRSA clearance, requires a more rigorous approach to support the authors' conclusions. Concerns about experimental approach and the lack of specific experimental detail from this part of the manuscript lessen enthusiasm for this otherwise interesting study.

Major Concerns:

1) In figure 4A and figure 5A, the authors used MitoTracker dye to define the association of mitochondria with MRSA phagosomes. Although this dye is designed to label the mitochondria, the dye appeared to be accumulating within phagosomes rather than juxtaposed with phagosomes. The confocal microscopy images were not of sufficiently high quality to unambiguously assess the mitochondrial staining. To confirm the MitoTracker data which is key to the study conclusions, the authors could use antibodies to stain mitochondrial markers post fixation. There are many commercially available validated antibodies that could be used to label mitochondria: MT-ND1 Monoclonal Antibody (18G12BC2) (Thermo Fisher Cat# 43-8800) and Tom20 Rabbit Antibody (Proteintech Cat#11802-1-AP).

2) Despite the authors' claim that they are using relatively low multiplicity of infection (5 or 20), the cells shown in the figures have a very high number of internalized bacteria, which leads to concerns that the cellular response may not be physiological. Key experiments should be repeated such that host cells have a lower number of bacteria (perhaps 3-5, rather than >20).

2) The authors quantified co-localization of the MRSA phagosome with MitoTracker to support their claim that loss of Caspase11 facilitates MRSA association with mitochondria. However, it is not clear how the authors quantified this co-localization in figure 4B and figure 5B. The authors should provide more detailed methods on how they performed the quantification in order to support their conclusion.

3) The authors propose that Caspase11 facilitates MRSA evasion of macrophage mROS-mediated killing. This argument is not very compelling because the only evidence supporting this hypothesis is that Antimycin A (Ant A) treatment decreased colony forming units (CFUs) only in Caspase11-deficient macrophages, but not Wild type macrophages. Treatment with Ant A increased mROS in Caspase11-deficient macrophages independent of infection. In addition, MRSA are being killed by both WT and Caspase11-deficient macrophages. This suggests that MRSA does not evade mROS killing, but rather Caspase11-deficient macrophages produce higher levels of mROS in response to Antimycin A stress, which happens to contribute to MRSA clearance. The authors should carefully rephrase their conclusions and the title of the manuscript in a way that reflects their data.

4) In figure 5C, the authors showed that macrophages treated with Cytochalasin D (Cyto D) or with Cyto D/Ant A had decreased CFU in both WT and Caspase11-deficient macrophages. Drug toxicity could be a factor in both the host and bacteria; the authors should assess whether these drugs affect MRSA axenic growth/survival. The authors also should assess the viability of macrophage when treated with these drugs with/without infection for the timeframe of the experiment.

5) The authors concluded that actin dynamics prevent MRSA phagosome association with

mitochondria in WT macrophages. Again, it is not clear how the authors quantified the co-localization of bacteria with Actin in figure 6 to support their conclusion. In addition, the difference between WT and Caspase11-deficient macrophages is subtle and macrophages are heavily infected.

6) In figure 4A and figure 6A, there are more bacteria associated with Caspase11 deficient macrophages compared with wild-type macrophages, albeit less live bacteria in Caspase11 deficient macrophages. Do Caspase11 deficient macrophages take up MRSA more efficiently when compared to WT macrophages? The authors should assess the role of Caspase11 in phagocytosis and provide an explanation for this phenomenon.

Minor concerns:

1) The authors should address the mechanism by which Gram-positive bacteria can induce Caspase11 activation, which could increase the impact of the manuscript. For example, is Caspase11 induced in response to fixed-bacterial (dead bacteria) infection? Recent work by Wolf and Reyes et al. (Cell 166, 624-636, July 28, 2016) described that Gram-positive bacterial peptidoglycan can be processed in the phagosomes into N-acetylglucosamine (NAG), which is released into the cytosol to activate the inflammasome. Is macrophage stimulation with peptidoglycan and targeting NAG to the cytosol induce Caspase11 dependent pro-inflammatory cytokines?

2) In lines 130-133, the authors concluded that Caspase11 contributes to secretion of IL-1 and Casp7 independently of Gasdermin D. The authors should perform statistical analysis post-test in figure 1B and 1C between Casp11^{-/-} and Gsdmd^{-/-} samples to test if the data support their conclusion.

3) The authors used the CRISPR/Cas9 system in figure 2 to genetically delete CASP4 and CASP5. However, they did not confirm whether these cells have lost expression of the proteins.

4) In line 218, the authors indicate that phagosome acidification is needed for MRSA replication. The authors have not shown that MRSA is replicating in phagosomes. One suggestion would be to change the word replication to survival.

5) Nomenclature of mRNA, protein, mice genotype and human cell genotype should be consistent. Examples: mouse mRNA and protein should be referred to as Casp11 and Casp11 respectively. Human mRNA and protein should be written as CASP4 and CASP4 respectively.

Referee #2:

GENERAL COMMENTS:

This study focuses on very novel biology, namely a regulatory role for CASP11 in curtailing macrophage antimicrobial responses against Gram-positive *S. aureus*. Since almost all of the existing literature on CASP11 in innate immunity has focused on recognition of cytosolic LPS from Gram-negative bacteria, this manuscript would undoubtedly be of great interest to the innate immunity community. The initial data in the manuscript on CASP11 dependence for macrophage responses to *S. aureus*, as well as the effect of CASP11 deficiency on bacterial loads in vivo and in vitro appear to be strong. However, I have a number of concerns about the methods, tools and interpretations relating to later mechanistic data sets in the manuscript (i.e. mitochondrial association, mtROS as the mechanism etc). Without additional controls and more appropriate tools, my view is that there are a number of major deficiencies in the manuscript as it stands. A particular concern is the almost exclusive use of pharmacological agents that will have broad impacts on cell functions (e.g Ant A, NAC, Cyto D, rapamycin, wortmannin), without the required controls and supporting evidence through alternate approaches (e.g. genetic). As it stands, the proposed model has not been robustly tested. If such issues can be carefully addressed and resolved, then the major findings in this manuscript are likely to be of broad interest to researchers in the innate immunity field.

SPECIFIC ISSUES:

1. Figure 1: this figure should also provide an analysis of cell death (e.g. LDH release) for the

various knock-out macrophage populations that were assessed (both control and infected). This would seem to be an important read-out to include, given the role of CASP11 in cell death responses triggered by Gram-negative bacteria. If there are differences in cell death between wild type and CASP11 k/o macrophages after *S. aureus* infection, then this would have major implications for interpreting subsequent data in the manuscript (e.g. CFU within macrophages) - this seems likely, given that CASP1 is activated in wild type cells but not CASP11 k/o cells. Was cell death assessed for experiments with mouse macrophages in Figure 1 (such data are provided for CASP4/5 in human cells in Figure 2)? Also, why are IL-1alpha and KC levels reduced in CASP11 k/o cells - is CASP11 mediating a transcriptional response or could this relate to differences in cell viability (noting that TNF is produced very rapidly, so this control is not really sufficient by itself)? This seems important for understanding these findings. A more minor point relates to the CASP7 cleavage data - is this necessary? This is not followed up on further, and so it seems like a distraction from the main story. If the authors wish to keep the data in this manuscript, perhaps it could be better integrated into the manuscript.

2. Figure 2: this figure should show that the various knock-out cell lines are indeed deficient in CASP4, 5 and/or 1 by immunoblotting. This is an important control that needs to be included.

3. Figure 3: compared to the in vivo effect in panel A (which is striking), the effect of CASP11 deficiency in reducing loads in macrophages (panel B) is much less pronounced (though still statistically significant). Assessment of an MOI range, or analysis of an earlier time point, in assessing intramacrophage CFU would solidify this aspect of the study. This seems important, since this panel provides the rationale for focusing on macrophages from this point on.

4. Figure 4 - I have several concerns with this Figure:

Panel A: the staining of cells with Mitotracker shows what appears to be mitochondrial staining, as well as staining that looks like it might be staining the *S. aureus* itself - for the overlap with USA300, I would not expect the Mitotracker staining pattern to be so perfectly spherical (even if there is extensive mitochondrial fission - hence my concern that the Mitotracker is directly staining the bacteria itself in this image). To address this issue, the authors could stain wild type and CASP11 k/o *S. aureus*-infected macrophages for a mitochondrial marker (e.g. Tom20) and/or use a genetically-encoded system in which mitochondria are specifically marked with a fluorescent protein (+/- CASP11 silencing), instead of relying exclusively on Mitotracker staining to reach their conclusions. This would ensure that these data are robust and appropriately interpreted i.e. that there is indeed enhanced colocalization of *S. aureus* with mitochondria in the absence of CASP11. I did not find the supporting EM data in Fig S1 to be overly convincing.

Panel C: NAC is used to inhibit mtROS in these experiments. This is a general antioxidant, and there are specific mtROS-quenching agents that would be much more appropriate to use in these experiments e.g. SS-31 or mitoQ. Also, this figure is very difficult to interpret - all data should be normalized to just the wild type control cells, and the wild type control and the CASP11 k/o control data should be plotted on this graph, so that one can see what the baseline level of mtROS for CASP11 k/o cells is versus the wild type cells. This information is crucial for interpreting these data. At present, it seems that the data has been normalized for both wild type and CASP11 k/o control cells, thus clouding interpretation of these data.

Panel D: BMM are treated with antimycin A (Ant A) for 24 h. I am concerned that this will affect BMM viability, affecting data interpretation. LDH release data or another read-out of cell death should accompany this figure, as without such data, this panel is uninterpretable. Earlier time points for CFU analysis would also be appropriate (e.g. 2.5h as per Fig 3B and possibly an intermediate time point e.g. 8h). Also, the statement relating to this figure "This indicates that enhanced proximity to mitochondria...." (bottom of page 8) is, in my opinion, far too over-reaching in its claims. At the very least, rescue experiments with SS-31 or mitoQ are required to at least show that this negates the Ant A effect (corresponding LDH/cell death data would be required for this as well).

5. Figure 5 - similar concerns to those above for Figure 4 apply to this figure, including (a) colocalization of mitochondria and *S. aureus* on the basis of Mitotracker staining only; and (b) the effect of Cyto D and Ant A+Cyto D on wild type and CASP11 k/o macrophage viability after *S. aureus* infection (CFU data are not interpretable without such data e.g. LDH release assays - it seems likely that the general health of the cells may be affected by 24 h co-treatment with Ant A +

Cyto D). Further, given that Cyto D will block bacterial uptake, it is crucial to know that the gentamicin exclusion is completely effective in the experiments i.e. there is no release of *S. aureus* and reinfection of macrophages - otherwise there would be major issues with interpreting the data. Have the authors plated culture supernatants at multiple time points after the 4 h time point to confirm that there are no viable bacteria extracellularly? Presumably there will not be, but given the nature of the experimental design, this possibility needs to be absolutely excluded.

6. In general, the mechanistic components in this study rely entirely on agents that will have numerous and broad effects on cells (Ant A, NAC, Cyto D, rapamycin, wortmannin). At least some complementary genetic approaches would be highly desirable to add weight to the conclusions that are drawn from the use of these rather blunt tools. For example, *S. aureus* mutants that are either resistant to ROS or display enhanced sensitivity to ROS could be used in wild type versus CASP11 k/o macrophages to confirm that the CFU differences relate to ROS-mediated killing. Alternatively, it is possible to genetically antagonize TLR-inducible mtROS production by various means - for example, ECSIT silencing in wild type versus CASP11 k/o macrophages could be used to determine if this selectively reduces mtROS and increases bacterial loads in CASP11 k/o macrophages. Such experiments would not offer any further mechanistic insight, but they would test the proposed model more robustly than the current pharmacological approaches do. CASP11 localization studies in control vs *S. aureus* infected macrophages would also be desirable, since the involvement of CASP11 in the entire response is based entirely on CASP11 k/o data (i.e. a single system).

7. The authors cite a recent study on the importance of mitoROS in the clearance of MRSA by macrophages (Abuaita et al, 2018; PMID: 30449314), but they do so in quite a limited way. Given how relevant this recent publication appears to be to this study, it would seem appropriate to provide further comment in the Discussion on the relevance of findings from the former study e.g. whether mitochondria-derived vesicles (MDVs) and Sod2 may be relevant. Also, there was really no discussion on possible mechanisms by which CASP11 might actually cause the separation of MRSA-containing vacuoles from mitochondria. Perhaps the Discussion could provide at least a brief analysis of possible mechanisms that might account for this phenotype.

MINOR:

8. All immunoblots should provide arrows and sizes corresponding to either molecular weight markers that were used OR show the sizes for all indicated bands e.g. the two pro-CASP11 bands in Figure 1A. This also applies to several other panels e.g. Figures 1B and 2A, where there are many banding patterns for CASP1 and no discussion of this. The banding pattern for CASP1 in Figure 2A looks unusual.

9. In Figure 1B, the reduction in IL-1beta levels in *Gsdmd* k/o cells in the immunoblot looks marginal by comparison to the wild type cells. Is this an appropriate representative blot to show here, given the quantitation in the panel at right? Also, there is no normalization control for the quantified data at right (i.e. if cell numbers are slightly different between the knock-out populations, this is not controlled for). Were MTT assays, or similar, performed on cells plated in parallel to ensure that the plating densities were similar for the various k/o cell populations?

10. In Figure 1A, there seems to be very faint bands corresponding to CASP11 in the CASP11 k/o cells - is this from cross-reactive bands or spillover between lanes? If the latter, it would be preferable to show a cleaner blot where there is no background CASP11 in the CASP11 k/o cells.

11. In Figure 3A, it would be preferable to not have the scale starting at 10e3 on the y-axis.

Referee #3:

This manuscript by Krause et al describes a novel role for caspase-11 during infection with Methicillin-resistant *Staphylococcus aureus* (MRSA). This report dissected in detail how MRSA actively avoid mitochondria to prevent killing by mitochondrial ROS. Interestingly, this study demonstrates that this "avoidance" requires caspase-11, and that this process requires the actin machinery. Overall, this manuscript is the first to describe the role of caspase-11 in MRSA infection

in relation to mitochondria location and function, and I only have a few minor points.

Minor points:

- The authors described the role of Gasdermin in cytokine production in response to MRSA but did not show if Gasdermin affects the persistence of MRSA.
- The role of caspase-11 in actin cytoskeleton during infection with MRSA is interesting but was not well dissected and stopped at the examination of phalloidin. The authors should investigate the ability of caspase-11 to regulate the actin machinery during MRSA infection further as they previously described in response to Gram negative bacteria.
- The TEM figure is essential and should be added to one of the main figures instead of supplemental figures.
- Loading controls for all westerns are needed including Fig. 1B and 2A.
- Figure 4A and 5A, the mitotracker can be pseudo-colored with brighter color as red for better visualization.
- Have the authors looked at the location of caspase-11 in relation to the MRSA vacuole? Did they examine the location of p-cofilin in relation to MRSA vacuole? It would be interesting if feasible.

1st Revision - authors' response

15 July 2019

Referee #1:

The manuscript by Krause et al. describes the role of Caspase11 in macrophage innate immunity against MRSA. The manuscript shows two major findings that occur independently of each other. First, Caspase11 expression is induced during MRSA infection and contributes to pro-inflammatory cytokine production (IL-1, IL-1 and Cxcl1/ KC). Caspase11 deficient macrophages produce less pro-inflammatory cytokines when compared to WT macrophages. Second, the authors suggest that loss of Caspase11 increases macrophage bactericidal activity by enhancing mitochondrial association with MRSA-containing phagosomes. Mitochondrial association with phagosomes and enhancing macrophage bactericidal activity has been previously described for Gram-negative bacteria (West, et al, Nature 2011). This study evaluates a potential role for Caspase11 in controlling this process. The authors propose that Caspase11 facilitates MRSA evasion through macrophage killing with mROS.

The manuscript is clearly written and the authors used a robust genetic approach to assess the role of Caspase11 in pro-inflammatory cytokine production during MRSA infection. Although the role of Caspase11 in activation of the non-canonical inflammasome has been previously been described within the context of other infection models, especially Gram-negative bacteria, this study establishes the importance of Caspase11 in non-canonical inflammasome activation during infection by MRSA, a pathogen of public health significance. The conclusions of this part of the study are well supported by the data. However, the second finding of the study, regarding the role of Caspase11 in mitochondria-mediated MRSA clearance, requires a more rigorous approach to support the authors' conclusions. Concerns about experimental approach and the lack of specific experimental detail from this part of the manuscript lessen enthusiasm for this otherwise interesting study.

We thank the reviewer for the time and effort in thoroughly reading our manuscript and appreciate the feedback.

Major Concerns:

1) In figure 4A and figure 5A, the authors used MitoTracker dye to define the association of mitochondria with MRSA phagosomes. Although this dye is designed to label the mitochondria, the dye appeared to be accumulating within phagosomes rather than juxtaposed with phagosomes. The confocal microscopy images were not of sufficiently high quality to unambiguously assess the mitochondrial staining. To confirm the MitoTracker data which is key to the study conclusions, the authors could use antibodies to stain mitochondrial markers post fixation. There are many commercially available validated antibodies that could be used to label mitochondria: MT-ND1

Monoclonal Antibody (18G12BC2) (Thermo Fisher Cat# 43-8800) and Tom20 Rabbit Antibody (Proteintech Cat#11802-1-AP).

In agreement with the reviewer's comment we included a Tom20 staining in Fig. 4B and Fig. 6B (former Fig.5).

2) Despite the authors' claim that they are using relatively low multiplicity of infection (5 or 20), the cells shown in the figures have a very high number of internalized bacteria, which leads to concerns that the cellular response may not be physiological. Key experiments should be repeated such that host cells have a lower number of bacteria (perhaps 3-5, rather than >20).

We apologize if it was not clear. All confocal experiments were originally done using MOI 5. MOI 20 was only used when measuring cytokine release to be able to measure adequate amounts of cytokines. However, to address the reviewer's concern, the MOI was reduced for the new Figures 4B and 6B showing Tom20 in WT and *casp11*^{-/-} macrophages.

2) The authors quantified co-localization of the MRSA phagosome with MitoTracker to support their claim that loss of Caspase11 facilitates MRSA association with mitochondria. However, it is not clear how the authors quantified this co-localization in figure 4B and figure 5B. The authors should provide more detailed methods on how they performed the quantification in order to support their conclusion.

Total bacteria were scored per image as well as bacteria co-localized with MitoTracker. Bacteria co-localized with MitoTracker were expressed in % of the total number of bacteria. >1000 bacteria per experiment were scored for Fig. 4A and 6A and >400 bacteria per experiment for Fig. 4B and 6B.

3) The authors propose that Caspase11 facilitates MRSA evasion of macrophage mROS-mediated killing. This argument is not very compelling because the only evidence supporting this hypothesis is that Antimycin A (Ant A) treatment decreased colony forming units (CFUs) only in Caspase11-deficient macrophages, but not Wild type macrophages. Treatment with Ant A increased mROS in Caspase11-deficient macrophages independent of infection. In addition, MRSA are being killed by both WT and Caspase11-deficient macrophages. This suggests that MRSA does not evade mROS killing, but rather Caspase11-deficient macrophages produce higher levels of mROS in response to Antimycin A stress, which happens to contribute to MRSA clearance. The authors should carefully rephrase their conclusions and the title of the manuscript in a way that reflects their data.

The normalization of the MitoSOX data in Fig. 5A was changed (request of reviewer 2), now showing normalization of all conditions to WT NT. Fig. 5A shows a significant increase in MitoSOX fluorescence in MRSA-infected *casp11*^{-/-} macrophages compared to WT. However, no difference could be observed between Ant A-treated WT and *casp11*^{-/-} macrophages either non-infected or infected with MRSA. Therefore, we conclude that enhanced proximity to mitochondria promotes mtROS-driven MRSA clearance in response to Ant A in *casp11*^{-/-} macrophages.

4) In figure 5C, the authors showed that macrophages treated with Cytochalasin D (Cyto D) or with Cyto D/Ant A had decreased CFU in both WT and Caspase11-deficient macrophages. Drug toxicity could be a factor in both the host and bacteria; the authors should assess whether these drugs affect MRSA axenic growth/survival. The authors also should assess the viability of macrophage when treated with these drugs with/without infection for the timeframe of the experiment.

We thank the reviewer for this comment. Supplementary Fig.2A and B show bacterial growth curves to test the toxicity of the drugs used in this study. Except for FCCP no direct bactericidal effect was observed. Fig. S2C shows the LDH assay to test the viability of macrophages when treated with Ant A and Cyto D. No cytotoxicity was observed in macrophages treated with Cyto D. Ant A induced ~20% of cell death, yet no difference could be observed between WT and *casp11*^{-/-} macrophages, indicating that Ant A-induced cytotoxicity is similar in both genotypes.

5) The authors concluded that actin dynamics prevent MRSA phagosome association with mitochondria in WT macrophages. Again, it is not clear how the authors quantified the co-localization of bacteria with Actin in figure 6 to support their conclusion. In addition, the difference between WT and Caspase11-deficient macrophages is subtle and macrophages are heavily infected.

Total bacteria were scored per image as well as bacteria co-localized with phalloidin. Bacteria co-localized with phalloidin were expressed in % of the total number of bacteria. >1000 bacteria per experiment were scored. We agree that the effect is only mild, which we have indicated in the results section.

6) In figure 4A and figure 6A, there are more bacteria associated with Caspase11 deficient macrophages compared with wild-type macrophages, albeit less live bacteria in Caspase11 deficient macrophages. Do Caspase11 deficient macrophages take up MRSA more efficiently when compared to WT macrophages? The authors should assess the role of Caspase11 in phagocytosis and provide an explanation for this phenomenon.

We apologize for the missing information. A significant difference in the intracellular CFU between WT and *casp11*^{-/-} macrophages was only observed after 24h. No significant difference at 2.5h and 6h was observed (Fig. 3B and S1D). In addition, we compared intracellular bacterial numbers at 1h post infection to assess uptake (Fig. S1C). Macrophages were infected for 30 minutes, washed 3x with PBS, and treated with 100 µg/mL gentamicin for another 30 minutes. No significant difference was found.

Minor concerns:

1) The authors should address the mechanism by which Gram-positive bacteria can induce Caspase11 activation, which could increase the impact of the manuscript. For example, is Caspase11 induced in response to fixed-bacterial (dead bacteria) infection? Recent work by Wolf and Reyes et al. (Cell 166, 624-636, July 28, 2016) described that Gram-positive bacterial peptidoglycan can be processed in the phagosomes into N-acetylglucosamine (NAG), which is released into the cytosol to activate the inflammasome. Is macrophage stimulation with peptidoglycan and targeting NAG to the cytosol, induce Caspase11 dependent pro-inflammatory cytokines?

It has been shown very recently that lipoteichoic acid (LTA) derived from the cell wall of Gram-positive bacteria activates CASP11 (Hara et al. 2018, doi: 10.1016/j.cell.2018.09.047). The study was referenced throughout the manuscript. While we agree that it would be interesting to further assess which components in addition to LTA from Gram-positive bacteria might activate CASP11, we feel this would be beyond the scope of this manuscript.

2) In lines 130-133, the authors concluded that Caspase11 contributes to secretion of IL-1 and Casp7 independently of Gasdermin D. The authors should perform statistical analysis post-test in figure 1B and 1C between *Casp11*^{-/-} and *Gsdmd*^{-/-} samples to test if the data support their conclusion.

No significant difference in IL-1β cleavage and secretion was found between *casp11*^{-/-} and *gsdmd*^{-/-} macrophages. The sentence in line 133-136 was rephrased to “Interestingly, *gsdmd*^{-/-} macrophages demonstrated mildly increased levels of secreted IL-1β and CASP7 than their *casp11*^{-/-} and *casp1*^{-/-} counterparts”.

3) The authors used the CRISPR/Cas9 system in figure 2 to genetically delete CASP4 and CASP5. However, they did not confirm whether these cells have lost expression of the proteins.

We apologize for the oversight. The generation of THP-1 cells lacking CASP4, CASP5, or both by Seth L. Masters has been previously published (doi: 10.1002/eji.201545655). The publication provides proof of lost protein expression. The reference was added to the manuscript in the corresponding results and material&methods sections.

4) In line 218, the authors indicate that phagosome acidification is needed for MRSA replication. The authors have not shown that MRSA is replicating in phagosomes. One suggestion would be to change the word replication to survival.

We thank the reviewer for this comment and changed “replication” to “survival”.

5) Nomenclature of mRNA, protein, mice genotype and human cell genotype should be consistent. Examples: mouse mRNA and protein should be referred to as Casp11 and Casp11 respectively. Human mRNA and protein should be written as CASP4 and CASP4 respectively.

We used the nomenclature according to MGI. Gene names are all italics. For mouse the first letter is capitalized (ex. *Casp4*) and for human the gene name is all uppercase (ex. *CASP4*). Protein names for mouse and human are all uppercase (ex. CASP4). Homozygous recessive mouse strains are written in all lower case and italics (ex. *casp1^{-/-}*).

Referee #2:

GENERAL COMMENTS:

This study focuses on very novel biology, namely a regulatory role for CASP11 in curtailing macrophage antimicrobial responses against Gram-positive *S. aureus*. Since almost all of the existing literature on CASP11 in innate immunity has focused on recognition of cytosolic LPS from Gram-negative bacteria, this manuscript would undoubtedly be of great interest to the innate immunity community. The initial data in the manuscript on CASP11 dependence for macrophage responses to *S. aureus*, as well as the effect of CASP11 deficiency on bacterial loads in vivo and in vitro appear to be strong. However, I have a number of concerns about the methods, tools and interpretations relating to later mechanistic data sets in the manuscript (i.e. mitochondrial association, mtROS as the mechanism etc). Without additional controls and more appropriate tools, my view is that there are a number of major deficiencies in the manuscript as it stands. A particular concern is the almost exclusive use of pharmacological agents that will have broad impacts on cell functions (e.g Ant A, NAC, Cyto D, rapamycin, wortmannin), without the required controls and supporting evidence through alternate approaches (e.g. genetic). As it stands, the proposed model has not been robustly tested. If such issues can be carefully addressed and resolved, then the major findings in this manuscript are likely to be of broad interest to researchers in the innate immunity field.

We thank the reviewer for the time and effort in thoroughly reading our manuscript and appreciate the feedback.

SPECIFIC ISSUES:

1. Figure 1: this figure should also provide an analysis of cell death (e.g. LDH release) for the various knock-out macrophage populations that were assessed (both control and infected). This would seem to be an important read-out to include, given the role of CASP11 in cell death responses triggered by Gram-negative bacteria. If there are differences in cell death between wild type and CASP11 k/o macrophages after *S. aureus* infection, then this would have major implications for interpreting subsequent data in the manuscript (e.g. CFU within macrophages) - this seems likely, given that CASP1 is activated in wild type cells but not CASP11 k/o cells. Was cell death assessed for experiments with mouse macrophages in Figure 1 (such data are provided for CASP4/5 in human cells in Figure 2)? Also, why are IL-1alpha and KC levels reduced in CASP11 k/o cells - is CASP11 mediating a transcriptional response or could this relate to differences in cell viability (noting that TNF is produced very rapidly, so this control is not really sufficient by itself)? This seems important for understanding these findings. A more minor point relates to the CASP7 cleavage data - is this necessary? This is not followed up on further, and so it seems like a distraction from the main story. If the authors wish to keep the data in this manuscript, perhaps it could be better integrated into the manuscript.

In agreement with the reviewer’s suggestion, we provided cell death data for Fig.1 (shown in Fig. S1B). Cells in Fig. 1 were infected with MOI 20 to be able to measure adequate amounts of cytokines. The corresponding LDH data show reduced LDH release in *casp11^{-/-}*, *gsdmd^{-/-}*, and *casp1^{-/-}* macrophages (line 130-133). However, in Fig. 3B, which shows the CFU data, cells were infected with MOI 5 and the corresponding LDH analysis (Fig. 3C) reveals no significant

difference in cell death between WT, *caspl1*^{-/-}, and *caspl1*^{-/-} macrophages at MOI 5 indicating that the difference in CFU between WT and *caspl1*^{-/-} macrophages is cell death independent. IL-1 α cleavage and secretion have been shown to be regulated by CASP11 in the literature (Casson *et al.* 2013, doi: 10.1371/journal.ppat.1003400, and Wiggins *et al.* 2019, doi: 10.1111/ace.12946). Both publications are cited in line 70.

With regard to KC/CXCL1, we previously reported decreased secretion of KC from *caspl1*^{-/-} macrophages infected with the Gram-negative pathogen *Burkholderia cenocepacia* (Krause *et al.* 2018, doi: 10.1080/15548627.2018.1491494). Here we also report reduced KC release from *caspl1*^{-/-} macrophages infected with MRSA indicating a general defect (line 137-140). However, it is currently not clear what mechanism lies behind this phenotype. Since it will require extensive experiments to answer this question, we will leave the study of the role of caspase-11 on KC secretion to another study by our group or others.

The CASP7 cleavage data were moved to the supplemental figures (Fig. S1A).

2. Figure 2: this figure should show that the various knock-out cell lines are indeed deficient in CASP4, 5 and/or 1 by immunoblotting. This is an important control that needs to be included.

We apologize for the oversight. The generation of THP-1 cells lacking CASP4, CASP5, or both by Seth L. Masters has been previously published (doi: 10.1002/eji.201545655). The publication provides proof of lost protein expression. The reference was added to the manuscript in the corresponding results and material&methods sections.

3. Figure 3: compared to the in vivo effect in panel A (which is striking), the effect of CASP11 deficiency in reducing loads in macrophages (panel B) is much less pronounced (though still statistically significant). Assessment of an MOI range, or analysis of an earlier time point, in assessing intramacrophage CFU would solidify this aspect of the study. This seems important, since this panel provides the rationale for focusing on macrophages from this point on.

We agree with the reviewer's comment and provided CFU data at MOI 0.5, 5, and 20 for 2.5h, 6h, and 24h (Fig.3B, Fig. S1D and E). Only the 24h time point shows a significant difference in the intracellular bacterial loads between WT and *caspl1*^{-/-} macrophages.

4. Figure 4 - I have several concerns with this Figure:

Panel A: the staining of cells with Mitotracker shows what appears to be mitochondrial staining, as well as staining that looks like it might be staining the *S. aureus* itself - for the overlap with USA300, I would not expect the Mitotracker staining pattern to be so perfectly spherical (even if there is extensive mitochondrial fission - hence my concern that the Mitotracker is directly staining the bacteria itself in this image). To address this issue, the authors could stain wild type and CASP11 k/o *S. aureus*-infected macrophages for a mitochondrial marker (e.g. Tom20) and/or use a genetically-encoded system in which mitochondria are specifically marked with a fluorescent protein (+/- CASP11 silencing), instead of relying exclusively on Mitotracker staining to reach their conclusions. This would ensure that these data are robust and appropriately interpreted i.e. that there is indeed enhanced colocalization of *S. aureus* with mitochondria in the absence of CASP11. I did not find the supporting EM data in Fig S1 to be overly convincing.

In agreement with the reviewer's suggestion, macrophages were stained with an antibody recognizing Tom20 and co-localization with MRSA was assessed. Fig. 4B shows increased co-localization of MRSA with Tom20 in *caspl1*^{-/-} macrophages.

Panel C: NAC is used to inhibit mtROS in these experiments. This is a general antioxidant, and there are specific mtROS-quenching agents that would be much more appropriate to use in these experiments e.g. SS-31 or mitoQ. Also, this figure is very difficult to interpret - all data should be normalized to just the wild type control cells, and the wild type control and the CASP11 k/o control data should be plotted on this graph, so that one can see what the baseline level of mtROS for CASP11 k/o cells is versus the wild type cells. This information is crucial for interpreting these data. At present, it seems that the data has been normalized for both wild type and CASP11 k/o control cells, thus clouding interpretation of these data.

In accordance to the reviewer's suggestion all conditions for each genotype were normalized to the WT non-treated control group (Fig. 5A). We also included the mtROS inhibitor MitoQ in this graph. The data show increased mtROS in *caspl1*^{-/-} macrophages infected with MRSA.

Panel D: BMM are treated with antimycin A (Ant A) for 24 h. I am concerned that this will affect BMM viability, affecting data interpretation. LDH release data or another read-out of cell death should accompany this figure, as without such data, this panel is uninterpretable. Earlier time points for CFU analysis would also be appropriate (e.g. 2.5h as per Fig 3B and possibly an intermediate time point e.g. 8h). Also, the statement relating to this figure "This indicates that enhanced proximity to mitochondria...." (bottom of page 8) is, in my opinion, far too over-reaching in its claims. At the very least, rescue experiments with SS-31 or mitoQ are required to at least show that this negates the Ant A effect (corresponding LDH/cell death data would be required for this as well).

We provided LDH data for Ant A in Fig. S2C. Ant A induced ~20% of cell death, yet no difference could be observed between WT and *caspl1*^{-/-} macrophages, indicating that Ant A-induced cytotoxicity is similar in both genotypes.

Treating macrophages with a combination of Ant A and a ROS scavenger such as MitoQ is not feasible since Ant A is a potent ROS inducer and the scavenger will be rapidly consumed. However, to address the reviewer's comment, we added CFU data of WT and *caspl1*^{-/-} macrophages treated with MitoQ alone (line 219-221). Fig. S2E shows that MitoQ significantly increases intracellular MRSA in *caspl1*^{-/-} macrophages.

5. Figure 5 - similar concerns to those above for Figure 4 apply to this figure, including (a) colocalization of mitochondria and *S. aureus* on the basis of Mitotracker staining only; and (b) the effect of Cyto D and Ant A+Cyto D on wild type and CASP11 k/o macrophage viability after *S. aureus* infection (CFU data are not interpretable without such data e.g. LDH release assays - it seems likely that the general health of the cells may be affected by 24 h co-treatment with Ant A + Cyto D). Further, given that Cyto D will block bacterial uptake, it is crucial to know that the gentamicin exclusion is completely effective in the experiments i.e. there is no release of *S. aureus* and reinfection of macrophages - otherwise there would be major issues with interpreting the data. Have the authors plated culture supernatants at multiple time points after the 4 h time point to confirm that there are no viable bacteria extracellularly? Presumably there will not be, but given the nature of the experimental design, this possibility needs to be absolutely excluded.

In agreement with the reviewer's concerns we provided the following data:

(a) Tom20 staining for WT cell treated with Cyto D in Fig.5B

(b) LDH assay for cells treated with Cyto D in Fig. S2C. The cytotoxicity in Cyto D treated cells was similar to the control group and no significant increase in cytotoxicity was observed with the combination Ant A/Cyto D.

(c) Since Cyto D blocks bacterial uptake, the treatment has been given after infection with MRSA. To further proof the effectiveness of gentamicin, we plated cell culture supernatants for either non-treated, Ant A-, Cyto D-, or Ant A/Cyto D-treated macrophages and determined extracellular bacterial loads. As shown in Fig. S2D only mild numbers of extracellular bacteria were present in gentamicin-treated cell culture supernatants. Furthermore, there was no significant difference in extracellular bacterial loads between non-treated, Ant A-, Cyto D-, or Ant A/Cyto D-treated WT and *caspl1*^{-/-} macrophages.

6. In general, the mechanistic components in this study rely entirely on agents that will have numerous and broad effects on cells (Ant A, NAC, Cyto D, rapamycin, wortmannin). At least some complementary genetic approaches would be highly desirable to add weight to the conclusions that are drawn from the use of these rather blunt tools. For example, *S. aureus* mutants that are either resistant to ROS or display enhanced sensitivity to ROS could be used in wild type versus CASP11 k/o macrophages to confirm that the CFU differences relate to ROS-mediated killing. Alternatively, it is possible to genetically antagonize TLR-inducible mtROS production by various means - for example, ECSIT silencing in wild type versus CASP11 k/o macrophages could be used to determine if this selectively reduces mtROS and increases bacterial loads in CASP11 k/o macrophages. Such experiments would not offer any further mechanistic insight, but they would test the proposed model more robustly than the current pharmacological approaches do. CASP11 localization studies in control vs *S. aureus* infected macrophages would also be desirable, since the involvement of CASP11 in the entire response is based entirely on CASP11 k/o data (i.e. a single system).

Unfortunately, the suggested MRSA mutants either resistant or more sensitive to ROS are not readily available and we feel that the generation of such mutants would be out of the scope of this manuscript. We hope that the cell death analyses we provided for the pharmacological agents we used will strengthen our conclusion.

We agree that CASP11 localization studies would be desirable, however due to the nature of the antibody (clone 17D9) these studies are not feasible at this point (the antibody is not suitable for immunofluorescence). We certainly tried and used *caspl1*^{-/-} macrophages as negative control, yet the signal was not specific.

7. The authors cite a recent study on the importance of mitoROS in the clearance of MRSA by macrophages (Abuaita et al, 2018; PMID: 30449314), but they do so in quite a limited way. Given how relevant this recent publication appears to be to this study, it would seem appropriate to provide further comment in the Discussion on the relevance of findings from the former study e.g. whether mitochondria-derived vesicles (MDVs) and Sod2 may be relevant. Also, there was really no discussion on possible mechanisms by which CASP11 might actually cause the separation of MRSA-containing vacuoles from mitochondria. Perhaps the Discussion could provide at least a brief analysis of possible mechanisms that might account for this phenotype.

In agreement with the reviewer's comment, we added a section to the discussion (line 380-386) commenting on the relevant findings by Abuaita et al. 2018. We also included a sentence regarding defective actin dynamics in *caspl1*^{-/-} macrophages as possible mechanism for decreased separation of the MRSA-containing vacuole from mitochondria (line 392-396).

MINOR:

8. All immunoblots should provide arrows and sizes corresponding to either molecular weight markers that were used OR show the sizes for all indicated bands e.g. the two pro-CASP11 bands in Figure 1A. This also applies to several other panels e.g. Figures 1B and 2A, where there are many banding patterns for CASP1 and no discussion of this. The banding pattern for CASP1 in Figure 2A looks unusual.

The corresponding sizes have been added to all Western Blots. We agree that the antibody for murine CASP1 reveals multiple bands which are also shown on the product sheet for this antibody (Cat# AG-20B-0042-C100, <https://adipogen.com/ag-20b-0042-anti-caspase-1-p20-mouse-mab-casper-1.html>). We added arrows to differentiate between pro-CASP1 and the cleavage product.

9. In Figure 1B, the reduction in IL-1beta levels in Gsdmd k/o cells in the immunoblot looks marginal by comparison to the wild type cells. Is this an appropriate representative blot to show here, given the quantitation in the panel at right? Also, there is no normalization control for the quantified data at right (i.e. if cell numbers are slightly different between the knock-out populations, this is not controlled for). Were MTT assays, or similar, performed on cells plated in parallel to ensure that the plating densities were similar for the various k/o cell populations?

The Western Blot for IL-1beta in Fig.1B is representative. To provide a loading control for cell culture supernatant Western Blots we loaded equal amounts of corresponding cell lysates from this experiment and probed for GAPDH.

10. In Figure 1A, there seems to be very faint bands corresponding to CASP11 in the CASP11 k/o cells - is this from cross-reactive bands or spillover between lanes? If the latter, it would be preferable to show a cleaner blot where there is no background CASP11 in the CASP11 k/o cells.

The samples shown on the Western Blot in Fig. 1A were re-run. We applied more vigorous blocking to provide a cleaner Western Blot.

11. In Figure 3A, it would be preferable to not have the scale starting at 10e3 on the y-axis.

The y-axis has been changed and is now starting at 0.

Referee #3:

This manuscript by Krause et al describes a novel role for caspase-11 during infection with Methicillin-resistant *Staphylococcus aureus* (MRSA). This report dissected in detail how MRSA actively avoid mitochondria to prevent killing by mitochondrial ROS. Interestingly, this study demonstrates that this "avoidance" requires caspase-11, and that this process requires the actin machinery. Overall, this manuscript is the first to describe the role of caspase-11 in MRSA infection in relation to mitochondria location and function, and I only have a few minor points.

We thank the reviewer for his/her positive comments and his support for the publication of our manuscript.

Minor points:

- The authors described the role of Gasdermin in cytokine production in response to MRSA but did not show if Gasdermin affects the persistence of MRSA.

In agreement with the reviewer's suggestion, we tested intracellular survival of MRSA in *gsdmd*^{-/-} compared to WT macrophages (Fig. 3E). We found no difference in intracellular survival between WT and *gsdmd*^{-/-} macrophages.

- The role of caspase-11 in actin cytoskeleton during infection with MRSA is interesting but was not well dissected and stopped at the examination of phalloidin. The authors should investigate the ability of caspase-11 to regulate the actin machinery during MRSA infection further as they previously described in response to Gram negative bacteria.

While we agree that it would be interesting to further assess the role of CASP11 in regulating the actin machinery, we feel this can be a focus for another manuscript. We provided additional immunofluorescence data (colocalization of MRSA with the mitochondrial protein Tom20) to strengthen our conclusion that MRSA is found in proximity to mitochondria in macrophages lacking CASP11.

- The TEM figure is essential and should be added to one of the main figures instead of supplemental figures.

The figure was moved to main figures (Fig. 4E).

- Loading controls for all westerns are needed including Fig. 1B and 2A.

To provide loading controls for cell culture supernatant Western Blots we loaded equal amounts of corresponding cell lysates and probed for GAPDH. The Western Blots showing GAPDH have been added to Fig. 1, 2, 5, and 6.

- Figure 4A and 5A, the mitotracker can be pseudo-colored with brighter color as red for better visualization.

We increased the brightness in Fig. 4A and 6A (former Fig. 5A).

- Have the authors looked at the location of caspase-11 in relation to the MRSA vacuole? Did they examine the location of p-cofilin in relation to MRSA vacuole? It would be interesting if feasible.

We agree that CASP11 localization studies would be interesting, however due to the nature of the antibody (clone 17D9) these studies are not feasible at this point. We certainly tried and used *casp11*^{-/-} macrophages as negative control, yet the signal was not specific. We did not look at p-cofilin since we wanted to focus the manuscript on the role of mtROS during MRSA infection.

Thank you for the submission of your revised manuscript to our editorial offices. We have now received the reports from the three referees that were asked to re-evaluate your study, you will find below. As you will see, the referees now support the publication of your manuscript in EMBO reports. However, referees #1 and #2 have some remaining concerns or further suggestions we ask you to address in a final revised version of your manuscript. Please also provide a detailed point-by-point response addressing these remaining concerns with the final revised manuscript.

Further, I have these editorial requests:

- Please add a conflict of interest statement below the author contributions.
- It seems author Arwa Abu Khweek is not mentioned in the author contributions. Please check and fix.
- It seems panels Fig. 5A and 5B are not called out in the text. Please check and fix.
- Please move the Materials and Methods section up that it follows Discussion and precedes the author contributions.
- The Expanded View format, which will be displayed in the main HTML of the paper in a collapsible format, has replaced the Supplementary information. Please follow the nomenclature Figure EV1, Figure EV2 etc. Thus, please rename the Figures S1-S3 into Figures EV1-3. Then please add the figure legend for these to the main manuscript document file in a section called Expanded View Figure Legends after the main Figure Legends section.
- Please define in all figure legends (main and EV) the nature of the replicates (biological vs. technical).
- For all boxplots shown, please define the horizontal bands within the boxes, the box ranges, the "+" symbols, and the error bars.
- Please add uniform scale bars to all microscopic images. Do not write on the scale bars, but provide the size information in the respective figure legend.
- Please supply an ORCID ID for the corresponding author. Please find instructions on how to link your ORCID ID to your account in our manuscript tracking system in our author guidelines: <http://www.embopress.org/page/journal/14693178/authorguide#authorshipguidelines>
- Please find attached a word file of the manuscript text (provided by our publisher) with changes we ask you to include in your final manuscript text, and some queries, we ask you to address. Please provide your final manuscript file with track changes, in order that we can see the modifications done.

In addition I would need from you:

- a short, two-sentence summary of the manuscript
- two to three bullet points highlighting the key findings of your study
- a schematic summary figure (in jpeg or tiff format with the exact width of 550 pixels and a height of not more than 400 pixels) that can be used as a visual synopsis on our website.

REFeree REPORTS

Referee #1:

This is a resubmitted manuscript by Krause et al. that describes how lack of Caspase11 enhances the ability of macrophages to kill MRSA by preventing dissociation of mitochondria from the vicinity of phagosomes. The paper also identifies that Caspase11 contributes to production of proinflammatory cytokines (IL-1 and IL-1) and chemokines (CXCL1/KC) during MRSA infection. The authors address most of the reviewers' concerns and suggestions. However, there are still concerns especially about the immunofluorescence data that are required to be addressed to support the authors' conclusions.

Minor Concerns:

- 1) The MitoTracker staining does not reflect general mitochondrial staining as observed by the new set of data using Tom20 antibody and the images from the MitoTracker experiment do not represent the quantified data. In figure 4A and figure 6A the macrophages are highly infected and the images do not represent the quantified data in figure 4C and figure 6C, respectively. For example, in figure 4A there are approximately 23 bacteria in Caspase11 deficient macrophages, of which 14 bacteria are positive for MitoTracker. This suggests that 60% of the bacteria are positive for MitoTracker and figure 4C shows only 10% of bacteria are positive for MitoTracker. I suggest replacing those images with more representative images, moving the data into a supplement figure or omitting the data as the new Tom20 staining data is sufficient to support the authors' conclusion.
- 2) The authors conclude that MRSA infection prevents mitochondrial recruitment to the vicinity (juxtaposition) of phagosomes. However, based on the Tom20 staining, there are Tom20 positive signals present adjacent to MRSA phagosomes in wild-type macrophages (figure 4B and figure 6B). This suggests that mitochondria are in juxtaposition to MRSA phagosomes, and that perhaps a lack of Caspase11 could increase this association. The authors should restate the conclusion in the abstract section to precisely reflect the data.
- 3) In figure 4B, Tom20 staining in Caspase11 deficient macrophages is dimmer than in wild-type macrophages. The authors should address why this is the case. Were these images taken at different z-sections of the cell, which could account for the lower Tom20 staining?
- 4) The scale bar for images within the same experiment should be constant throughout the manuscript.

----- Referee #2:

The authors have made a concerted effort to address many of the issues that I had raised in my review of the original manuscript - some of the additional data sets that are provided in the revised manuscript certainly alleviate some of my major concerns (e.g. showing that there is minimal cell death when using low MOI's of *S. aureus*, thus alleviating concerns about the impact of cell death on interpretation of intracellular bacterial load data). However, the major issue around the very extensive use of pharmacological agents without complementary approaches (e.g. genetic) still remains. Therefore, I am not completely convinced of the proposed model/mechanism - it may well be correct, but the data to support the model are largely phenomenological, with functional data relying on non-specific pharmacological approaches. In addition, the effects of mitoQ in antagonizing the effect of Casp11 deficiency are really quite modest - this suggests that there is more going on than just mitochondrial ROS or that these experiments haven't been optimised (compare effect of Casp11 deficiency vs level of rescue by MitoQ). I think some textual modifications are required throughout the manuscript to ensure that the authors don't over-interpret their data (e.g. see below). There are also a number of other minor issues below that the authors should address. All of this said, the findings are certainly novel and will be of great interest to the infection and immunity research community.

SPECIFIC ISSUES:

1. Given the lack of complementary functional data that causally links mitochondrial ROS to the antimicrobial effects of Casp11 deficiency (e.g. genetic data to support pharmacological data), the authors should carefully edit their manuscript to ensure that the presented data are not over-

interpreted. This comment is also made in light of the fact that (1) the appropriate replotting of mitochondrial ROS data (now Fig. 5A) means that these data are not particularly compelling (NAC and mitoQ don't reduce mitochondrial ROS levels); and (2) the new data showing the effect of mitoQ on bacterial loads in Casp11-deficient macrophages in Fig. S2E is really quite modest (albeit statistically significant). In my view, in the absence of more compelling functional data, there are several statements throughout the manuscript (even the title) that are over-reaching in their claims and should be modified accordingly e.g. pg 9 "This indicates that enhanced proximity to mitochondria is necessary to eliminate...."; pg 15 "...macrophages deficient of CASP11, resulting in mtROS-mediated bacterial clearance". This is also true for other conclusions that are based on fairly non-specific pharmacological agents e.g. pg 11 "...wortmannin significantly reduced the intracellular burden of MRSA, indicating that the suppression of autophagy facilitates MRSA clearance from murine macrophages" (wortmannin has many more effects on cell function, apart from its effects on autophagy). In summary, a number of textual modifications are required to ensure that the authors are not so definitive in their claims.

2. The MOI range experiments (Fig. S1E) are provided in response to one of my original queries. The MOI 0.5 data in this figure are certainly useful. However, given the cell death observed at MOI 20 (Fig. S1B), I don't think that the MOI 20 data in Fig. S1E can be readily interpreted. Therefore, I recommend removing the right-hand side panel (MOI 20) from Fig. S1E i.e. only showing the left-hand side panel (MOI 0.5).

3. Fig. S2A - it would be preferable to present these data using a linear scale, rather than logarithmic. It is very difficult to see if there are any minor (but potentially important) effects on growth with the data presented in this way.

4. pg 8, last line: should be Fig. 5A (not 4C).

5. pg 9, line 216: should be Fig. 5B (not 4D).

6. Fig 5A legend should indicate how long cells were infected for, prior to mitochondrial ROS levels being assessed.

7. pg 9-10: Fig. S3A is described before Fig. S2F (should present figure panels in the order in which the data are described).

Referee #3:

The authors have addressed all the concerns raised by the reviewers.

2nd Revision - authors' response

16 September 2019

Referee #1:

This is a resubmitted manuscript by Krause et al. that describes how lack of Caspase11 enhances the ability of macrophages to kill MRSA by preventing dissociation of mitochondria from the vicinity of phagosomes. The paper also identifies that Caspase11 contributes to production of proinflammatory cytokines (IL-1 and IL-1) and chemokines (CXCL1/KC) during MRSA infection. The authors address most of the reviewers' concerns and suggestions. However, there are still concerns especially about the immunofluorescence data that are required to be addressed to support the authors' conclusions.

Minor Concerns:

1) The MitoTracker staining does not reflect general mitochondrial staining as observed by the new set of data using Tom20 antibody and the images from the MitoTracker experiment do not represent the quantified data. In figure 4A and figure 6A the macrophages are highly infected and the images do not represent the quantified data in figure 4C and figure 6C, respectively. For example, in figure 4A there are approximately 23 bacteria in Caspase11 deficient macrophages, of which 14 bacteria are positive for MitoTracker. This suggests that 60% of the bacteria are positive for MitoTracker and figure 4C shows only 10% of bacteria are positive for MitoTracker. I suggest replacing those images with more representative images, moving the data into a supplement figure or omitting the data as the new Tom20 staining data is sufficient to support the authors' conclusion.

We thank the reviewer for this great suggestion. Confocal Images in Figure 4A and and 6A have been replaced by images with lower bacterial numbers to better reflect the quantified data in Figure 4C and 6C. For better visualization the images were pseudo-colored in red.

2) The authors conclude that MRSA infection prevents mitochondrial recruitment to the vicinity (juxtaposition) of phagosomes. However, based on the Tom20 staining, there are Tom20 positive signals present adjacent to MRSA phagosomes in wild-type macrophages (figure 4B and figure 6B). This suggests that mitochondria are in juxtaposition to MRSA phagosomes, and that perhaps a lack of Caspase11 could increase this association. The authors should restate the conclusion in the abstract section to precisely reflect the data.

We thank the reviewer for this comment. Although there is Tom20 staining present adjacent to MRSA in WT macrophages, there is significantly decreased co-localization of MRSA with Tom20 in WT compared to *casp11*^{-/-} macrophages (Fig. 4B and D) suggesting there is a greater distance between mitochondria and MRSA in WT macrophages. Our TEM images show more mitochondria surrounding MRSA phagosomes in the absence of caspase-11. We agree that there is very mild co-localization with Tom20 (or MitoTracker) in WT macrophages, yet the majority of bacteria display no co-localization suggesting that MRSA evades mitochondrial association (also suggested by Cohen *et al.* 2018, <https://doi.org/10.1016/j.celrep.2018.02.027>). In agreement with the reviewer's suggestion, textual modifications were done in the abstract section.

3) In figure 4B, Tom20 staining in Caspase11 deficient macrophages is dimmer than in wild-type macrophages. The authors should address why this is the case. Were these images taken at different z-sections of the cell, which could account for the lower Tom20 staining?

Pictures were taken with same laser settings for WT and *casp11*^{-/-} macrophages. While the immunofluorescence intensity can differ between different areas on the cover slip, we didn't find any overall difference in Tom20 intensity between WT and *casp11*^{-/-} macrophages.

4) The scale bar for images within the same experiment should be constant throughout the manuscript.

Uniform scale bars were added to all microscopic images.

Referee #2:

The authors have made a concerted effort to address many of the issues that I had raised in my review of the original manuscript - some of the additional data sets that are provided in the revised manuscript certainly alleviate some of my major concerns (e.g. showing that there is minimal cell death when using low MOI's of *S. aureus*, thus alleviating concerns about the impact of cell death on interpretation of intracellular bacterial load data). However, the major issue around the very extensive use of pharmacological agents without complementary approaches (e.g. genetic) still remains. Therefore, I am not completely convinced of the proposed model/mechanism - it may well

be correct, but the data to support the model are largely phenomenological, with functional data relying on non-specific pharmacological approaches. In addition, the effects of mitoQ in antagonizing the effect of Casp11 deficiency are really quite modest - this suggests that there is more going on than just mitochondrial ROS or that these experiments haven't been optimised (compare effect of Casp11 deficiency vs level of rescue by MitoQ). I think some textual modifications are required throughout the manuscript to ensure that the authors don't over-interpret their data (e.g. see below). There are also a number of other minor issues below that the authors should address. All of this said, the findings are certainly novel and will be of great interest to the infection and immunity research community.

SPECIFIC ISSUES:

1. Given the lack of complementary functional data that causally links mitochondrial ROS to the antimicrobial effects of Casp11 deficiency (e.g. genetic data to support pharmacological data), the authors should carefully edit their manuscript to ensure that the presented data are not over-interpreted. This comment is also made in light of the fact that (1) the appropriate replotting of mitochondrial ROS data (now Fig. 5A) means that these data are not particularly compelling (NAC and mitoQ don't reduce mitochondrial ROS levels); and (2) the new data showing the effect of mitoQ on bacterial loads in Casp11-deficient macrophages in Fig. S2E is really quite modest (albeit statistically significant). In my view, in the absence of more compelling functional data, there are several statements throughout the manuscript (even the title) that are over-reaching in their claims and should be modified accordingly e.g. pg 9 "This indicates that enhanced proximity to mitochondria is necessary to eliminate..."; pg 15 "...macrophages deficient of CASP11, resulting in mtROS-mediated bacterial clearance". This is also true for other conclusions that are based on fairly non-specific pharmacological agents e.g. pg 11 "...wortmannin significantly reduced the intracellular burden of MRSA, indicating that the suppression of autophagy facilitates MRSA clearance from murine macrophages" (wortmannin has many more effects on cell function, apart from its effects on autophagy). In summary, a number of textual modifications are required to ensure that the authors are not so definitive in their claims.

We appreciate the concerns raised by the reviewer. We would like to clarify that we intentionally used several complementary pharmacological inhibitors to support each conclusion. We also included appropriate controls and tested the effect of each pharmacological agent on bacterial cultures to exclude direct anti-bacterial effects. This approach corroborated each finding. Nevertheless, in response to the reviewer's concerns, textual modifications were made throughout the manuscript according to the reviewer's recommendations.

2. The MOI range experiments (Fig. S1E) are provided in response to one of my original queries. The MOI 0.5 data in this figure are certainly useful. However, given the cell death observed at MOI 20 (Fig. S1B), I don't think that the MOI 20 data in Fig. S1E can be readily interpreted. Therefore, I recommend removing the right-hand side panel (MOI 20) from Fig. S1E i.e. only showing the left-hand side panel (MOI 0.5).

We thank the reviewer for his/her comment. We would like to clarify that Figure EV1E aims to show the MOI range. While there is a difference in LDH release at MOI 20 (Fig. EV1B), we showed that Gentamicin treatment is still very effective (Fig. EV2D) at 24 hours post infection preventing extracellular bacterial replication.

3. Fig. S2A - it would be preferable to present these data using a linear scale, rather than logarithmic. It is very difficult to see if there are any minor (but potentially important) effects on growth with the data presented in this way.

We changed the figures to a linear scale and also added the statistical analysis.

4. pg 8, last line: should be Fig. 5A (not 4C).

We apologize for the oversight and changed Fig. 4C to Fig. 5A.

5. pg 9, line 216: should be Fig. 5B (not 4D).

We apologize for the oversight and changed Fig. 4D to Fig. 5B.

6. Fig 5A legend should indicate how long cells were infected for, prior to mitochondrial ROS levels being assessed.

In response to the reviewer's comment, the information was added to the figure legend.

7. pg 9-10: Fig. S3A is described before Fig. S2F (should present figure panels in the order in which the data are described).

In response to the reviewer's comment now former Fig. S2F is Fig EV3B.

Referee #3:

The authors have addressed all the concerns raised by the reviewers.

We thank the reviewer for the positive feedback.

Corresponding Author Name: Amal O. Amer

Journal Submitted to: Embo Reports

Manuscript Number: EMBOR-2019-48109